# The *Drosophila* HNF4 nuclear receptor promotes glucose-stimulated insulin secretion and mitochondrial function in adults

**William E Barry, Carl S Thummel\***

Department of Human Genetics, University of Utah School of Medicine, Salt Lake City, United States

**Abstract** Although mutations in *HNF4A* were identified as the cause of Maturity Onset Diabetes of the Young 1 (MODY1) two decades ago, the mechanisms by which this nuclear receptor regulates glucose homeostasis remain unclear. Here we report that loss of *Drosophila HNF4* recapitulates hallmark symptoms of MODY1, including adult-onset hyperglycemia, glucose intolerance and impaired glucose-stimulated insulin secretion (GSIS). These defects are linked to a role for dHNF4 in promoting mitochondrial function as well as the expression of *Hex-C*, a homolog of the MODY2 gene *Glucokinase*. dHNF4 is required in the fat body and insulin-producing cells to maintain glucose homeostasis by supporting a developmental switch toward oxidative phosphorylation and GSIS at the transition to adulthood. These findings establish an animal model for MODY1 and define a developmental reprogramming of metabolism to support the energetic needs of the mature animal.

**\*For correspondence:** carl. thummel@genetics.utah.edu

**Competing interests:** The authors declare that no competing interests exist.

## Introduction

The global rise in the prevalence of diabetes has prompted increased efforts to advance our understanding of metabolic systems and how they become disrupted in the diseased state. Although genetics and environment have a significant impact on diabetes susceptibility, severity, and care, the causal factors are often complex and unclear. Several cases of familial diabetes have been identified, however, that show clear patterns of heritability due to monogenic disease alleles, highlighting these genes as critical factors for glycemic control. To date, mutations in 13 genes have been shown to cause autosomal dominant inheritance of Maturity Onset Diabetes of the Young (MODY1-13), representing the most common forms of monogenic diabetes. MODY patients typically present with hyperglycemia and impaired glucose-stimulated insulin secretion (GSIS) by young adulthood, while having normal body weight and lacking β-cell autoimmunity (*Fajans and Bell, 2011*). Consistent with this, several genes associated with MODY have well-characterized functions in glucose homeostasis, including the glycolytic enzyme *Glucokinase* (GCK/MODY2), and *Insulin* (INS/MODY10). Mechanistic insight into the anti-diabetic roles of other MODY genes, however, remains limited.

The genetic basis for the first MODY subtype was reported two decades ago, identifying loss-of-function mutations in *Hepatocyte Nuclear Factor 4A* (*HNF4A*) as responsible for MODY1 (*Yamagata et al., 1996*). HNF4A is a member of the nuclear receptor superfamily of ligand-regulated transcription factors, which play important roles in the regulation of growth, development, and metabolic homeostasis. Studies in mice demonstrated a critical requirement for *Hnf4A* in early development, with null mutants dying during embryogenesis due to defects in gastrulation (*Chen et al., 1994*). Heterozygotes, however, show no apparent phenotypes. As a result, tissue-specific genetic

**eLife digest** Diabetes is a complex disease that is caused by a combination of factors, including the person's habits and environment, as well as their genetic make-up. However, there are some rare forms of diabetes that are caused simply by mutations in single genes and are directly inherited. For example, it has been known for twenty years that a type of diabetes called "Maturity Onset Diabetes of the Young type 1" (or MODY1 for short) occurs when a gene called *HNF4* is mutated or deleted. The symptoms of MODY1 usually appear during early adulthood and include abnormally high levels of sugar in the blood, as well as the pancreas not being able to release the hormone insulin properly in response to these sugars.

Previous studies in mice have tried to understand how losing the HNF4 gene leads to MODY1. However, these mouse models did not fully recreate the symptoms of this disorder and the precise role of HNF4 in preventing diabetes remains unclear. Barry and Thummel have now used the fruit fly, because it is a model organism with simple genetics, to help shed light on this question. Furthermore, flies and mammals use many of the same pathways to control metabolism, making the fly a good model for the disease in humans.

Barry and Thummel deleted the *HNF4* gene in fruit flies and observed that the flies had all the symptoms that are typical in people with MODY1. These symptoms included high sugar levels and decreased production of insulin-like hormones. The experiments also showed that *HNF4* normally supports the proper expression of another gene called *Hex-C*; this gene encodes a protein that senses how much sugar is available and helps to keep the amount of sugar circulating the body within normal levels. Barry and Thummel went on to discover that the *HNF4* gene is required for the expression of some genes in structures called mitochondria, which provide most of the energy used by animal cells. Lastly, the *HNF4* gene became more active as the flies matured, and appeared to help the metabolism of a developing fruit fly transition towards that of an adult.

Together these findings show that *HNF4* protects against MODY1 by influencing several components of sugar metabolism in fruit flies. In the future, more studies are needed to understand how exactly *HNF4* acts in mitochondria and to explore if similar results are seen in mammals.

studies were used to investigate the functions of *Hnf4A* in key tissues where it is expressed, including the liver, kidney, intestine, and pancreatic β-cells. Two groups generated adult mice deficient for *Hnf4A* in β-cells with the goal of modeling MODY1 (*Gupta et al., 2005*; *Miura et al., 2006*). Although both studies reported impaired glucose tolerance in *Hnf4A* deficient mice, along with defects in GSIS, neither study observed sustained hypoinsulinemic hyperglycemia – the defining symptom that brings MODY1 patients to the clinic. As a result, we still have a limited understanding of the mechanisms by which *Hnf4A* maintains carbohydrate homeostasis and the molecular basis for MODY1.

Studies in *Drosophila* have revealed a high degree of conservation with major pathways that regulate cellular metabolism and systemic physiology in humans (*Diop and Bodmer, 2015*; *Owusu-Ansah and Perrimon, 2014*; *Padmanabha and Baker, 2014*; *Teleman et al., 2012*). This includes a central role for the insulin-signaling pathway in maintaining proper levels of circulating sugars through nutrient-responsive secretion of *Drosophila* insulin-like peptides (DILPs) from neuroendocrine cells in the fly brain (*Nassel et al., 2015*). Destruction of these insulin-producing cells (IPCs) results in elevated levels of circulating sugars, analogous to type 1 diabetes (*Rulifson et al., 2002*). In addition, nutrient-sensing mechanisms for insulin release are conserved in adult *Drosophila*, including roles for the Glut1 glucose transporter, mitochondrial metabolism, and ATP-sensitive potassium channels in the IPCs, which respond to the anti-diabetic sulfonylurea drug glibenclamide (*Fridell et al., 2009*; *Kreneisz et al., 2010*; *Park et al., 2014*). Consistent with these similarities, an increasing number of studies in *Drosophila* have proven relevant to mammalian insulin signaling and metabolic homeostasis, highlighting the potential to provide insight into human metabolic disorders such as diabetes (*Alfa et al., 2015*; *Owusu-Ansah and Perrimon, 2014*; *Park et al., 2014*; *Ugrankar et al., 2015*; *Xu et al., 2012*).

Here we describe our functional studies of *Drosophila* HNF4 (dHNF4) with the goal of defining its roles in maintaining carbohydrate homeostasis. dHNF4 is a close ortholog of human HNF4A, with 89% amino acid identity in the DNA-binding domain and 61% identity in the ligand-binding domain. The spatial expression patterns of the fly and mammalian receptors are also conserved through evolution, raising the possibility that they share regulatory activities (*Palanker et al., 2009*). In support of this, our previous studies of *dHNF4* mutant larvae demonstrated a critical role in fatty acid catabolism, leading to defects in lipid homeostasis that are similar to those caused by liver-specific *HNF4A* deficiency in mammals (*Palanker et al., 2009*). Here we report the first functional study of *dHNF4* mutants at the adult stage of development. Our studies show that adult *dHNF4* mutants display the hallmark symptoms of MODY1, including hyperglycemia, glucose intolerance and impaired GSIS. Metabolomic analysis of *dHNF4* mutants revealed coordinated changes in metabolites that are indicative of diabetes, along with an unexpected effect on mitochondrial activity. This was further evident in our RNA-seq and ChIP-seq studies, which indicate that *dHNF4* is required for the proper transcription of both nuclear and mitochondrial genes involved in oxidative phosphorylation (OXPHOS). A homolog of mammalian *GCK, Hex-C*, is also under-expressed in mutants. *dHNF4* appears to act through these pathways to promote GSIS in the IPCs and glucose clearance by the fat body. In addition, we show that *dHNF4* expression increases dramatically at the onset of adulthood, along with its downstream transcriptional programs. These studies suggest that *dHNF4* triggers a developmental transition that establishes the metabolic state of the adult fly, promoting GSIS and OXPHOS to support the energetic needs of the mature animal.

## Results

### *dHNF4* mutants are sugar intolerant and display hallmarks of diabetes

All genetic studies used a transheterozygous combination of *dHNF4* null alleles (*dHNF4^Δ17^/ dHNF4^Δ33^*) and genetically-matched controls that were transheterozygous for precise excisions of the EP2449 and KG08976 P-elements, as described previously (*Palanker et al., 2009*). Consistent with this earlier study, *dHNF4* null mutants die as young adults, with most mutants failing to emerge properly from the pupal case when raised under standard lab conditions (*Figure 1A*) (*Palanker et al., 2009*). While testing for potential dietary effects on *dHNF4* mutant viability, we discovered that sugar levels have a dramatic influence on their survival. When reared on either standard cornmeal food or a medium containing 15% sugar (2:1 glucose to sucrose, 8% yeast), less than 30% of mutant animals survive though eclosion, and the rest die primarily during the first day of adulthood (*Figure 1A,B*). In contrast, a five-fold reduction in dietary sugar content is sufficient to rescue most *dHNF4* mutants through eclosion and allow them to survive as adults for several weeks (*Figure 1B,C*). Sugar intolerance persists through adulthood, indicating that *dHNF4* plays a critical role in carbohydrate metabolism at this stage (*Figure 1C*). Notably, this dietary response is specific to alterations in carbohydrate levels, as calorically matched changes in dietary protein did not affect mutant viability (*Figure 1—figure supplement 1*).

To examine the effects of sugar consumption on the metabolic state of *dHNF4* mutants, major metabolites were measured in adult males raised on the low 3% sugar diet and transferred to the 3%, 9% or 15% sugar diet for three days. Although *dHNF4* mutants display elevated levels of triglycerides, similar to our observations in mutant larvae, these levels are not affected by the different sugar diets (*Figure 1—figure supplement 2A*). Similarly, while *dHNF4* mutants have reduced glycogen stores and a modest decrease in total protein, the severity of these phenotypes does not correlate with the improved viability due to decreasing dietary sugar (*Figure 1—figure supplement 2B, C*). In contrast, the abundance of free glucose is greatly elevated in *dHNF4* mutants on the 15% sugar diet, but is progressively reduced in mutants exposed to decreasing amounts of dietary sugar, similar to the response of diabetics to a low carbohydrate diet (*Figure 1D*). As expected, the accumulation of free glucose in *dHNF4* mutants represents increased levels in circulation and is accompanied by elevated levels of the glucose disaccharide trehalose (*Figure 1E,F*). Taken together, these results demonstrate that *Drosophila* HNF4 is required for proper glycemic control.

To assess whether the hyperglycemia in *dHNF4* mutants arises due to impaired glucose clearance, adult flies were subjected to an oral glucose tolerance test. Control and mutant animals were reared on the low sugar diet, fasted overnight, transferred to a glucose diet for one hour, and then

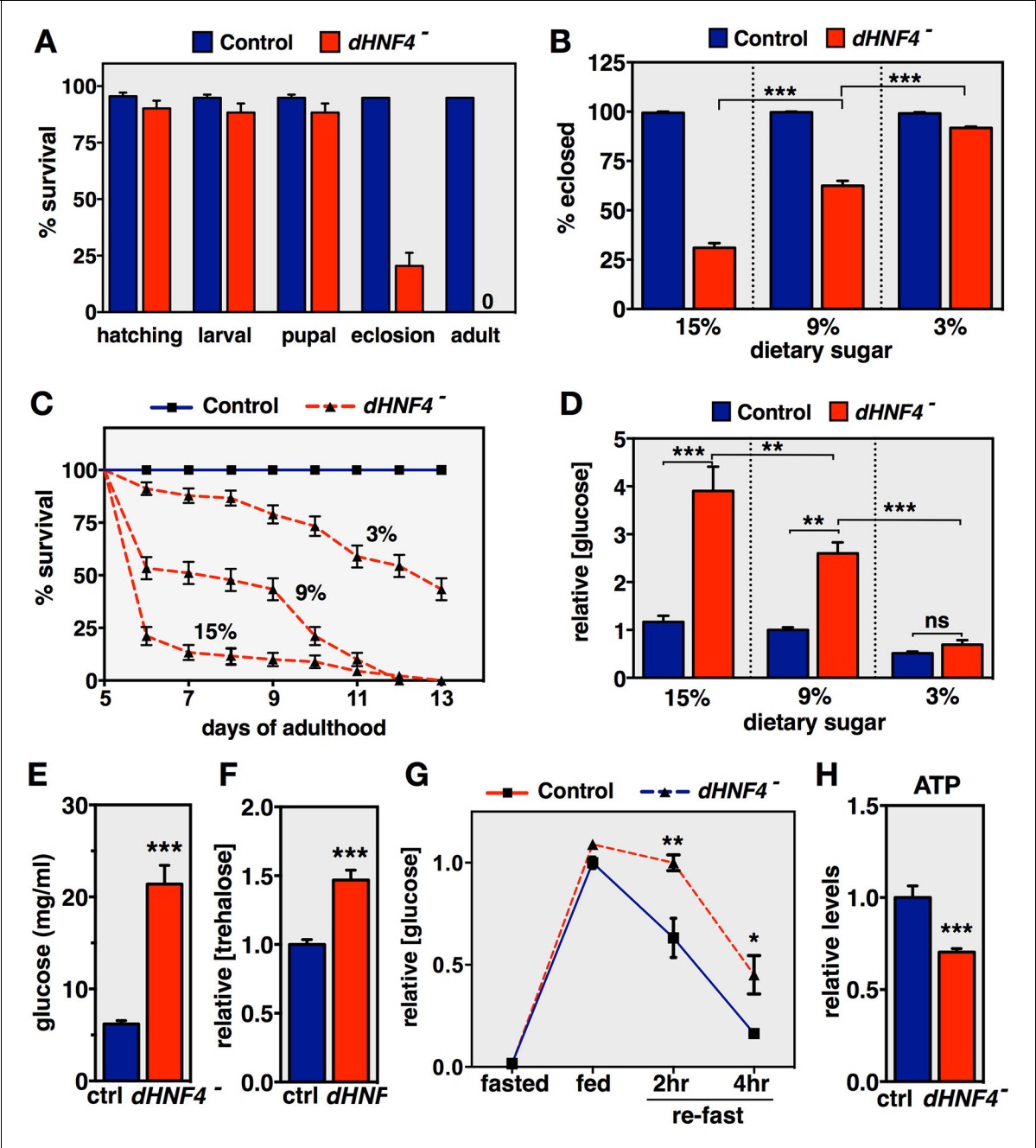

**Figure 1.** *dHNF4* mutants are sugar intolerant and display hallmarks of diabetes. (**A**) Percent survival of genetically-matched controls and *dHNF4* mutants at each stage of development when raised on standard media. Adult viability represents survival past the first day of adulthood. (**B**) Percent of control and *dHNF4* mutants that successfully eclose when reared on the 15%, 9%, or 3% sugar diet. (**C**) Controls and *dHNF4* mutants were reared on the 3% sugar diet until 5 days of adulthood, transferred to the indicated diet, and scored for survival. (**D**) Free glucose levels measured from whole animal lysates of controls and *dHNF4* mutants raised on the 3% sugar diet and transferred to the indicated diet for three days. (**E**) Circulating free glucose levels were measured from hemolymph extracted from control and *dHNF4* mutant adults raised on the 3% sugar diet and transferred to the 15% sugar diet for 1 day prior to analysis. (**F**) Trehalose levels measured from whole animal lysates of controls and *dHNF4* mutants raised on the 3% sugar diet and transferred to the 15% sugar diet for three days. (**G**) Oral-glucose tolerance test performed on adults raised on the 3% sugar diet, fasted overnight, fed on 15% glucose media for 1 hr, and then re-fasted for either 2 or 4 hr. Data represents relative free glucose levels from whole animal homogenates. (**H**) Relative ATP levels in control and *dHNF4* mutant adults raised on the 3% sugar diet and transferred to sugar-only medium (10% sucrose) for 1 day prior to analysis. Data is plotted as the mean ± SEM. ***$p \leq 0.001$, **$p \leq 0.01$, *$p \leq 0.05$.

The following figure supplements are available for figure 1:

*Figure 1 continued on next page*

*Figure 1 continued*

**Figure supplement 1.** Dietary sugar, but not protein, correlates with reduced *dHNF4* mutant survival.
**Figure supplement 2.** Profiling of major metabolites in *dHNF4* mutant adults fed different levels of dietary sugar.

re-fasted for 2 or 4 hr. Although *dHNF4* mutants display a normal postprandial spike in free glucose levels after feeding, glucose clearance is significantly impaired in mutant animals at both 2 and 4 hr, indicating glucose intolerance (*Figure 1G*). Taken together, these data demonstrate that *dHNF4* mutant adults display hallmarks of diabetes and may provide an animal model of MODY1.

## *dHNF4* mutants display defects in glycolysis and mitochondrial metabolism

Small-molecule gas chromatography/mass spectrometry (GC/MS) metabolomic analysis was used to further characterize the metabolic state of *dHNF4* mutants fed a 3% or 15% sugar diet (*Figure 2*). This study confirmed and extended our observations of their diabetic phenotype and revealed underlying defects in glucose homeostasis that are independent of dietary sugar content. Consistent with hyperglycemia, *dHNF4* mutants accumulate glycolytic metabolites on both diets. These include elevated glucose-6-phosphate, dihydroxyacetone phosphate (DHAP), and serine, which is produced from 3-phosphoglycerate, although the increased DHAP was only observed on the 15% sugar diet (*Figure 2*). Several other glucose-derived metabolites are aberrantly increased in *dHNF4* mutants, including sorbitol and fructose, which are intermediates in the polyol pathway (*Figure 2* and *Figure 2—figure supplement 1*). This pathway provides an alternate route for cellular glucose uptake under conditions of sustained hyperglycemia. As a result, these metabolites can accumulate to high levels in diabetics and correlate with neuropathy and nephropathy (*Gabbay, 1975*). *dHNF4* mutants also display increased levels of inosine, adenine, xanthine, hypoxanthine, and uric acid, which are purine metabolites that are associated with increased diabetes risk and diabetic nephropathy (*Figure 2*) (*Johnson et al., 2013*). Taken together, these findings reveal additional similarities between the *dHNF4* mutant phenotype and the metabolic complications of diabetes in humans. Finally, in addition to elevated carbohydrates, we observed increased levels of pyruvate and lactate accompanied by decreased levels of ATP, suggesting a potential defect in mitochondrial respiration (*Figure 1H*, *2*).

To further assess mitochondrial metabolism, *dHNF4* mutant adults were maintained on 10% sucrose medium for three days and analyzed for TCA cycle intermediates using GC/MS metabolomics. This approach was aimed at restricting the ability of dietary amino acids to replenish TCA cycle intermediates by anapleurosis to provide more robust detection of underlying defects in this pathway. Interestingly, *dHNF4* mutants display specific alterations in these metabolites, with increased abundance of citrate, aconitate, isocitrate, fumarate and malate, along with decreased levels of alpha-ketoglutarate and succinate, suggesting a specific block in TCA cycle progression (*Figure 3—figure supplement 1*). Taken together, these metabolite changes suggest that mitochondrial function is impaired in *dHNF4* mutants, providing a possible primary cause for their glucose intolerance.

## dHNF4 regulates nuclear and mitochondrial gene expression

As a first step toward identifying transcriptional targets of dHNF4 that mediate its effects on glucose homeostasis, we performed RNA-seq profiling in control and mutant adults. A total of 1370 genes are differentially expressed in *dHNF4* mutants ($\geq$1.5-fold change, 1% FDR), with just over half of these genes showing reduced abundance (726 down, 644 up) (*Supplementary file 1*). Gene ontology analysis revealed that the majority of the down-regulated genes correspond to metabolic functions, with the most significant category corresponding to oxidoreductases (*Supplementary file 2*). In contrast, the up-regulated genes largely correspond to the innate immune response, reflecting a possible inflammatory response in *dHNF4* mutants (*Supplementary file 2*). Interestingly, most of the transcripts encoded by the mitochondrial genome (mtDNA) are expressed at greatly reduced levels in mutant animals (*Supplementary file 1*). In *Drosophila*, as in humans, the mitochondrial genome contains 13 protein-coding genes, all of which encode critical components of the electron transport chain (ETC) that contribute to oxidative phosphorylation (OXPHOS). Further analysis of these

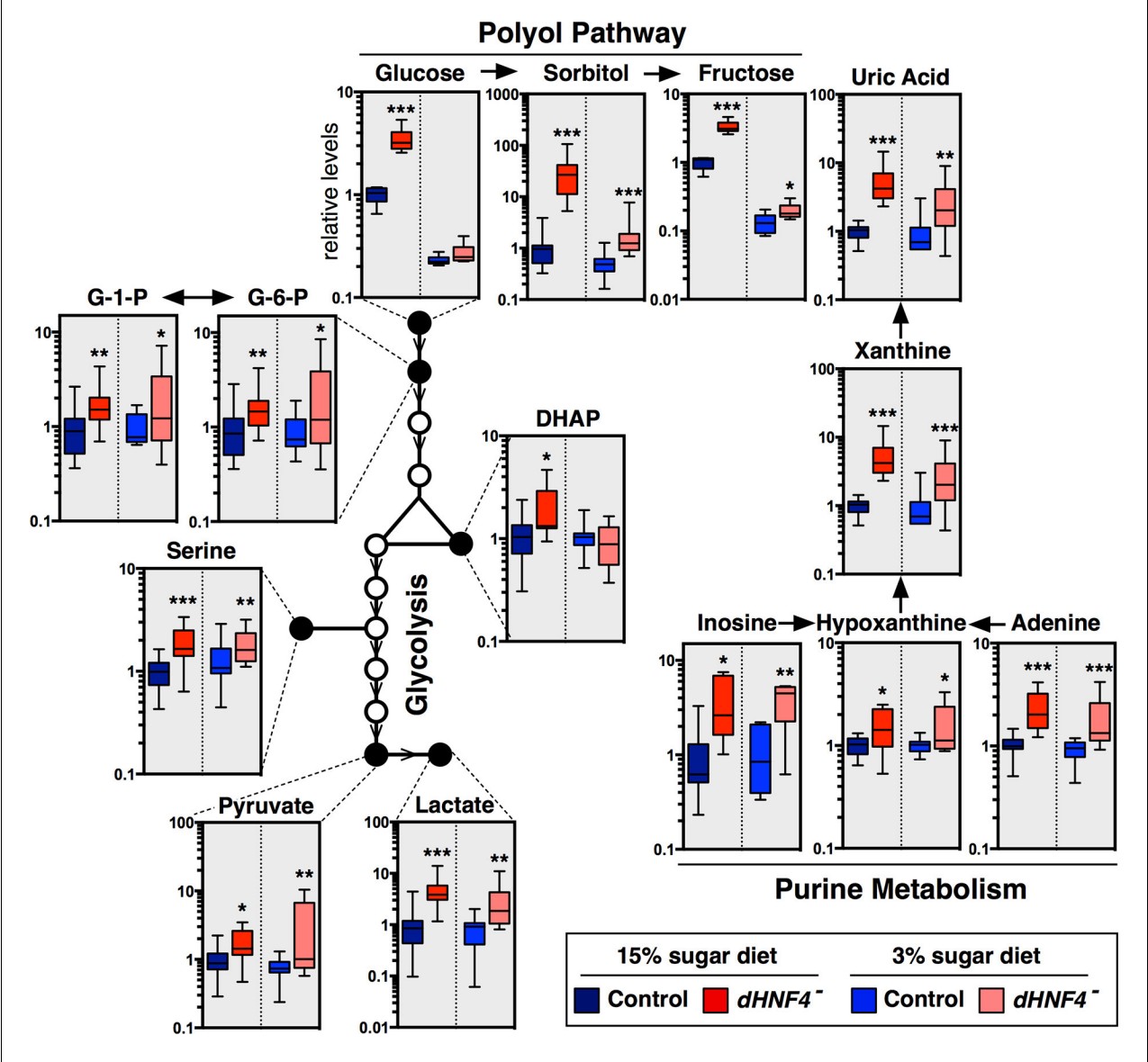

**Figure 2.** *dHNF4* mutants display defects in glycolysis and mitochondrial metabolism. GC/MS metabolomic profiling of controls and *dHNF4* mutants raised to adulthood on the 3% sugar diet, transferred to the indicated diet for 3 days, and subjected to analysis. Data were obtained from three independent experiments consisting of 5–6 biological replicates per condition and values were normalized to control levels on the 15% sugar diet. Box plots are presented on a log scale, with the box representing the lower and upper quartiles, the horizontal line representing the median, and the error bars representing the minimum and maximum data points. ***p≤0.001, **p≤0.01, *p≤0.05.

The following figure supplement is available for figure 2:

**Figure supplement 1.** *dHNF4* mutants show broad defects in carbohydrate homeostasis.

transcripts by northern blot hybridization confirmed their reduced expression in *dHNF4* mutants, corresponding to mtDNA genes involved in Complex I (*mt:ND1, mt:ND2, mt:ND4, mt:ND5*), Complex IV (*mt:Cox1, mt:Cox2, mt:Cox3*) and Complex V/ATP synthase (*mt:ATPase6, mt:ATPase8*), along with reduced levels of the mitochondrial large ribosomal RNA (*mt:lrRNA*) (*Figure 3A*, *Supplementary file 1*). Importantly, not all mtDNA genes are misregulated, as the expression of *mt: Cyt-b* is consistently unaltered in mutants (*Figure 3A*). In addition, the copy number of mtDNA is

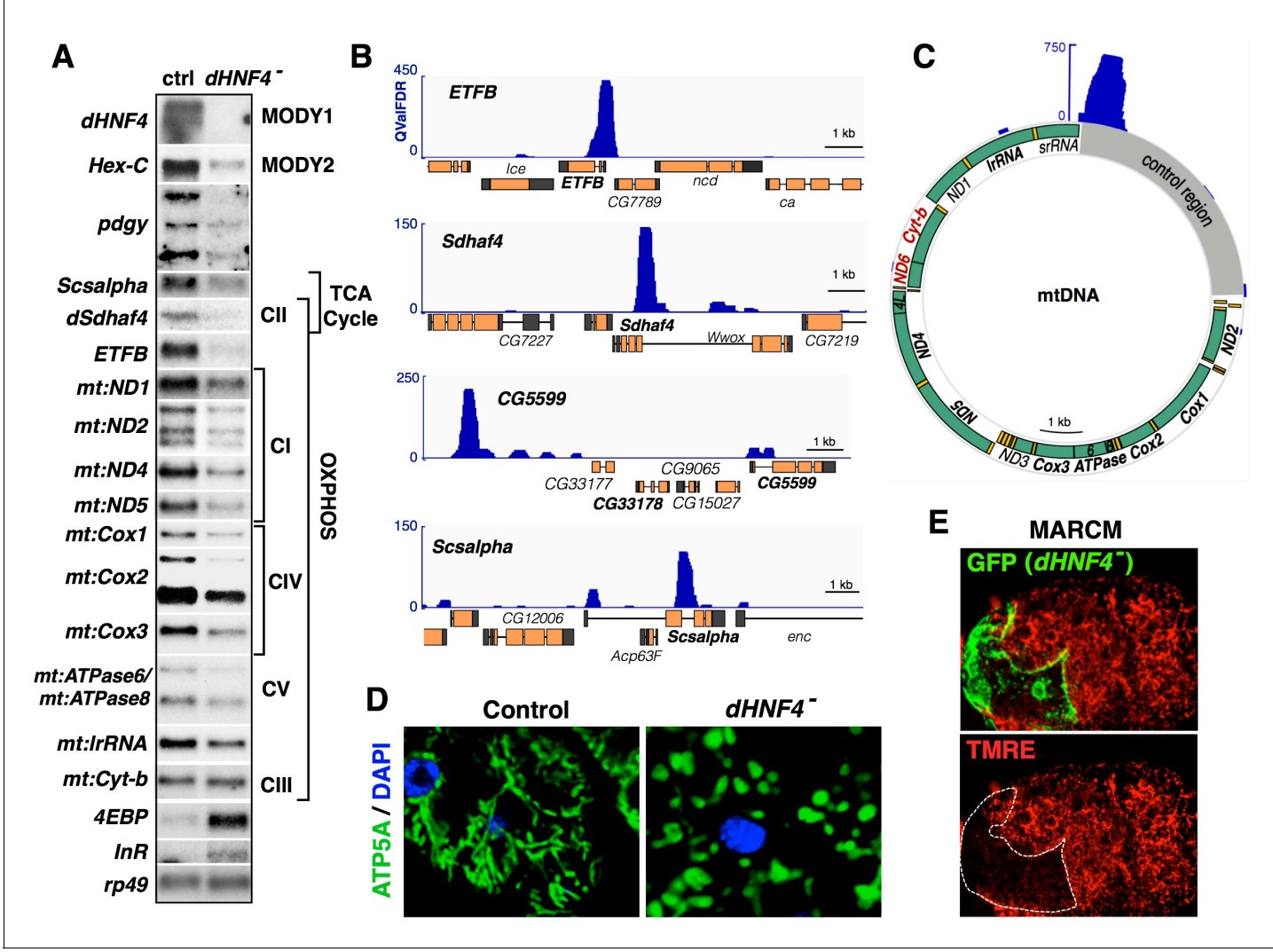

**Figure 3.** dHNF4 regulates nuclear and mitochondrial gene expression. (**A**) Validation of RNA-seq data by northern blot using total RNA extracted from control and *dHNF4* mutant adults. Affected transcripts include those involved in glucose homeostasis (*Hex-C, pdgy*), the electron transport chain (*Sdhaf4, mt:ND1, mt:ND2, mt:ND4, mt:ND5, mt:CoxI, mt:Cox2, mt:Cox3, mt:ATPase6/8, mt:Cyt-b* and *mt:lrRNA*), the TCA cycle (*Scsalpha, dSdhaf4*), and insulin signaling (*4EBP, InR*). *rp49* is included as a control for loading and transfer. Mitochondrial-encoded transcripts are indicated by the prefix '*mt*'. Depicted results were consistent across multiple experiments. (**B–C**) ChIP-seq analysis performed on adult flies for endogenous dHNF4 genomic binding shows direct association with both nuclear (**B**) and mitochondrial-encoded (**C**) genes involved in OXPHOS. Data tracks display *q* value FDR (QValFDR) significance values (*y*-axis) compared to input control, where QValFDR 50 corresponds to $P=10^{-5}$ and 100 corresponds to $P=10^{-10}$. Gene names in bold represent those expressed at reduced levels in *dHNF4* mutants by RNA-seq and/or northern blot analysis. Gene names in red (*ND6, Cyt-B*) denote the mtDNA-encoded transcriptional unit confirmed to show no change in *dHNF4* mutants. (**D**) Whole-mount immunostaining of adult fat body tissue for ATP5A (green) to detect mitochondria and DAPI (blue) to mark nuclei, showing fragmented mitochondrial morphology in *dHNF4* mutants. (**E**) Analysis of *dHNF4* mutant MARCM clones (GFP+) shows reduced mitochondrial membrane potential by TMRE staining of live fat body tissue from adult flies maintained on the 15% sugar diet.

The following figure supplements are available for figure 3:

**Figure supplement 1.** *dHNF4* mutants display changes in TCA cycle intermediates that correlate with changes in gene expression.

**Figure supplement 2.** *dHNF4* mutants display mitochondrial defects.

**Figure supplement 3.** Predicted functions of dHNF4 target genes.

unaffected in *dHNF4* mutants, suggesting that mitochondrial abundance is normal in these animals (*Figure 3—figure supplement 2A*).

Several nuclear-encoded OXPHOS genes also require *dHNF4* for their maximal expression, including genes that encode the alpha and beta subunits of the electron transfer flavoprotein (*ETFA* and *ETFB*), ETF-ubiquinone oxidoreductase (*ETF-QO*), and the Complex II (succinate dehydrogenase, SDH) assembly factor *dSdhaf4* (*Figure 3A*, *Supplementary file 1*). Similar to flies lacking *dSdhaf4*, *dHNF4* mutants display reduced steady-state levels of SDH complex as assayed by western blot (*Figure 3—figure supplement 1*) (*Van Vranken et al., 2014*). These observations are thus consistent with impaired mitochondrial SDH function, and suggest that *dSdhaf4* is a critical functional target of dHNF4. Additional genes involved in the TCA cycle are misexpressed in *dHNF4* mutants, including *Succinyl-CoA synthetase alpha (Scsalpha)*, *CG5599* (which encodes a protein with homology to the E2 subunit of the α-ketoglutarate dehydrogenase complex (α-KGDHC) as well as the E2 subunit of the branched-chain alpha-ketoacid dehydrogenase complex), *CG1544* (which encodes a homolog of α-KGDHC E1), as well as *Isocitrate dehydrogenase (IDH*, NADP$^+$-dependent) (*Figure 3A*, *Supplementary file 1*). These changes in gene expression are consistent with the observed changes in the levels of TCA cycle intermediates in *dHNF4* mutants, suggesting that they are functionally relevant to the mutant metabolic phenotype (*Figure 3—figure supplement 1*).

Notably, *dHNF4* mutants also have decreased expression of the *GCK* homolog *Hexokinase-C (Hex-C)* (*Figure 3A*). GCK is a tissue-specific glycolytic enzyme that is required for glucose sensing by pancreatic β-cells and glucose clearance by the liver. These activities, combined with the association of *GCK* mutations with MODY2, make *Hex-C* a candidate for mediating the effects of *dHNF4* on carbohydrate metabolism. The glucose transporter *CG1213* is also down-regulated in *dHNF4* mutants, along with *phosphoglucomutase (pgm)*, which is involved in glycogen metabolism, and *transaldolase* and *CG17333*, which are involved in the pentose phosphate shunt. Additionally, the gluconeogenesis genes *Pyruvate carboxylase (CG1514)* and *Phosphoenolpyruvate carboxykinase (Pepck, CG17725)* show reduced expression in mutant animals, similar to their dependence on *Hnf4A* for expression in the mammalian liver (*Supplementary file 1*) (*Chavalit et al., 2013*; *Yoon et al., 2001*). Finally, *dHNF4* mutants display transcriptional signatures of reduced insulin signaling, including up-regulation of the dFOXO-target genes *4EBP* and *InR* (*Figure 3A*, *Supplementary file 1*). Taken together, these findings indicate an important role for dHNF4 in mitochondrial OXPHOS and glucose metabolism, and suggest that it acts through multiple pathways to maintain glycemic control.

Chromatin immunoprecipitation followed by high-throughput sequencing (ChIP-seq) was performed to identify direct transcriptional targets of the receptor. Through this analysis, forty-seven genes were identified as high confidence targets by fitting the criteria of showing proximal dHNF4 binding along with reduced transcript abundance in mutant animals ($\geq$1.5 fold change, 1% FDR) (*Figure 3—figure supplement 3*, *Supplementary file 3*). These include nuclear-encoded OXPHOS genes such as *ETFB, ETF-QO, dSdhaf4,* and genes that encode TCA cycle factors *Scsalpha* and *CG5599* (*Figure 3B*). We also observed abundant and specific binding of dHNF4 within the control region of the mitochondrial genome (*Figure 3C* and *Supplementary file 3*). Taken together with our other results, these data suggest that dHNF4 is required to maintain normal mitochondrial function. Consistent with this, mitochondrial morphology is severely fragmented in mutant animals, and MARCM clonal analysis in the adult fat body shows reduced mitochondrial membrane potential in *dHNF4* mutant cells (*Figure 3D,E* and *Figure 3—figure supplement 2B,C*). In contrast, we were unable to detect changes in reactive oxygen species (ROS) in *dHNF4* mutant clones by DHE staining (*Figure 3—figure supplement 2D*). This might be due to the decreased levels of ROS-generating ETC complexes in *dHNF4* mutants, along with no detectable effect on the transcripts that encode ROS-scavenging enzymes, such as catalase and SOD (*Supplementary file 1*). Taken together, these data support the model that dHNF4 regulates both nuclear and mitochondrial gene expression to promote OXPHOS and maintain mitochondrial integrity.

## dHNF4 acts through multiple tissues and pathways to control glucose homeostasis

Tissue-specific RNAi was used to disrupt *dHNF4* expression in the IPCs, fat body, and intestine to examine the contributions of *dHNF4* in these tissues to systemic glucose homeostasis. This revealed a requirement in both the IPCs and fat body for glucose homeostasis, consistent with the well-

established roles of these tissues in insulin signaling and the regulation of circulating sugar levels (*Figure 4A*, *Figure 4—figure supplement 1B*). Our initial functional analysis of *dHNF4* target genes supports these tissue-specific activities and provides insights into the molecular mechanisms of *dHNF4* action. Tissue-specific inactivation of *Hex-C* by RNAi demonstrates that it is required in the fat body, but not the IPCs, to maintain normal levels of circulating glucose (*Figure 4B,C*). This is consistent with the important role of mammalian *GCK* for glucose clearance by the liver as well as its association with MODY2 (*Postic et al., 1999*). In contrast, both fat body and IPC-specific RNAi for the direct target of dHNF4, *CG5599*, significantly impaired glucose homeostasis (*Figure 4B,C*). This indicates that *CG5599* is required in each of these tissues for glycemic control, similar to dHNF4, suggesting that it is a key downstream target of the receptor. Although technical limitations prevent us from performing tissue-specific RNAi studies of mitochondrial-encoded transcripts, disruption of ETC Complex I by targeting a critical assembly factor, CIA30 (Complex I intermediate-associated protein 30 kDa) (*Cho et al., 2012*) in either the IPCs or fat body produced elevated levels of free

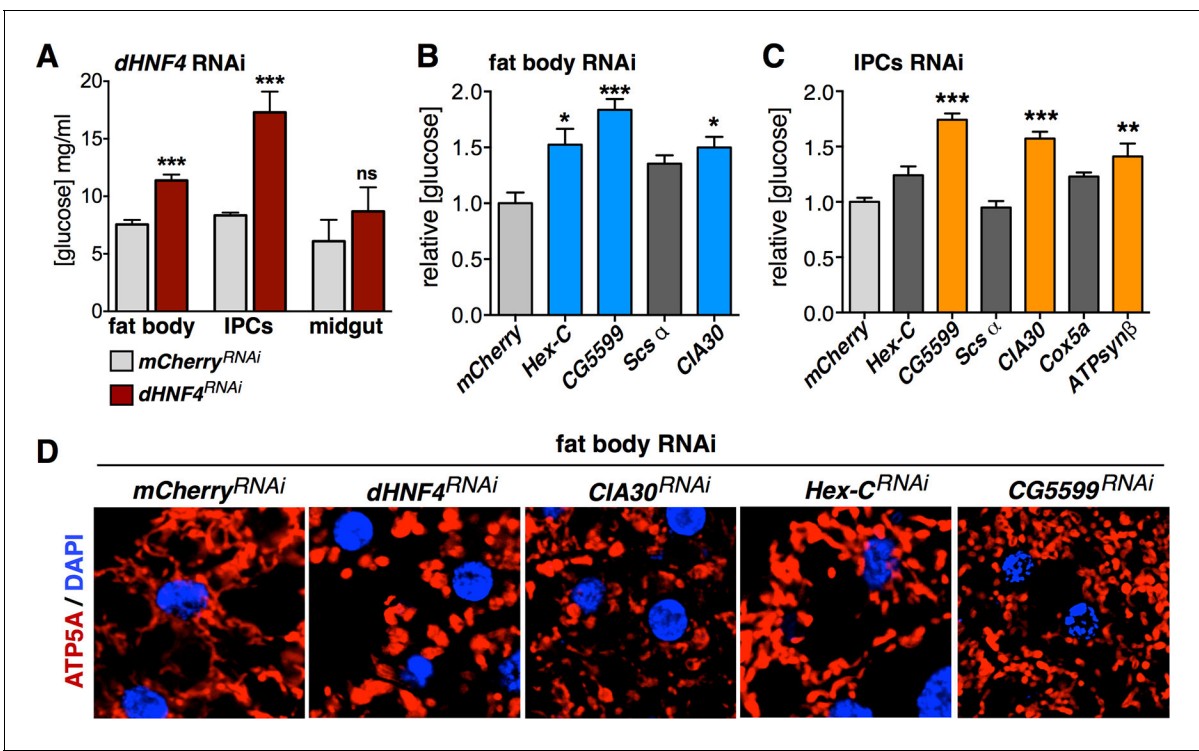

**Figure 4.** dHNF4 acts through multiple tissues and pathways to control glucose homeostasis. (**A**) Circulating glucose levels in adult males expressing tissue-specific RNAi against *mCherry* (TRiP 35785, grey bars) or *dHNF4* (TRiP 29375, dark red bars) in the fat body (*r4-GAL4*), IPCs (*dilp2-GAL4*), or midgut (*mex-GAL4*). (**B–C**) Relative free glucose levels in adult males on the 15% sugar diet expressing fat body (*r4-GAL4*, B) or IPC (*dilp2-GAL4*, C)-specific RNAi compared to *mCherry* RNAi controls (light grey bars). RNAi lines directed against *Hex-C*, *CG5599*, *Scsα*, *CIA30*, *Cox5a*, and *ATPsynβ* were obtained from the TRiP RNAi collection. Blue and orange bars depict significant changes in glucose levels. Dark grey bars are not significant. Data represents the mean ± SEM. \*\*\*p≤0.001, \*\*p≤0.01, \*p≤0.05. (**D**) Confocal imaging of mitochondrial morphology (marked by ATP5A immunostaining, red) in the adult fat body from animals expressing fat-body specific RNAi (*r4-GAL4*). The extended network of mitochondria seen in controls is disrupted and appears more punctate upon RNAi for *dHNF4*, *CIA30*, or *CG5599*, indicative of mitochondrial fragmentation. No effect is seen upon RNAi for *Hex-C*.

The following figure supplements are available for figure 4:

**Figure supplement 1.** *dHNF4* is required in the insulin-producing cells and fat body to maintain glucose homeostasis.

**Figure supplement 2.** Fat body-specific disruption of the electron transport chain causes sugar intolerance.

**Figure supplement 3.** Additional RNAi lines confirming the importance of *Hex-C* in the fat body for glycemic control.

glucose (*Figure 4B,C*). Fat body-specific RNAi for *dHNF4, CIA30, or CG5599* resulted in fragmented mitochondrial morphology, consistent with previous reports of CIA30 loss of function and the onset of mitochondrial dysfunction (*Figure 4D*) (*Cho et al., 2012*). In contrast, RNAi for *Hex-C* had no detectable effect on mitochondrial morphology (*Figure 4D*).

While RNAi for Complex V (*ATPsynβ* RNAi) in the fat body caused lethality prior to eclosion, IPC-specific RNAi produced viable adults that appeared normal but displayed significant hyperglycemia (*Figure 4C*). In contrast, disruption of ETC Complex IV in the IPCs (*Cox5a* RNAi) failed to produce hyperglycemia, while RNAi in the fat body caused lethality prior to adulthood, similar to *ATPsynβ*. This premature lethality was accompanied by severe developmental delay and more than 50% of the animals dying prior to puparium formation when raised on the 15% sugar diet. Interestingly, we discovered that these animals are sugar intolerant, similar to *dHNF4* mutants, such that rearing them on the 3% sugar diet allowed for 100% survival to puparium formation while also alleviating the developmental delay (*Figure 4—figure supplement 2*). Although adult viability was not achievable through dietary intervention, these findings demonstrate that ETC function in the fat body is important for sugar tolerance during development, similar to the requirement for dHNF4. Taken together, these data reveal important roles for *dHNF4* in both the IPCs and fat body to maintain glucose homeostasis, likely in part by promoting *Hex-C* expression in the fat body for glucose clearance and supporting mitochondrial function and OXPHOS in both the IPCs and fat body.

## *dHNF4* is required for glucose-stimulated DILP2 secretion

The requirement for *dHNF4* function in the IPCs for systemic glucose homeostasis fits with the important roles for *Hnf4A* in mouse pancreatic β-cells as well as the contribution of β-cell physiology to the onset of MODY1. Accordingly, we examined if *dHNF4* mutants display defects in GSIS. We used an experimental approach developed for this purpose in *Drosophila* larvae, assaying for the steady-state levels of DILP2 peptide in the IPCs using a fasting/refeeding paradigm (*Geminard et al., 2009*). As expected, DILP2 accumulates in the IPCs of fasted control animals and is effectively released into circulation in response to glucose feeding (*Figure 5A,B*). In contrast, while DILP2 accumulates normally in fasted *dHNF4* mutants, it fails to respond to dietary glucose stimulation, despite these animals having normal IPC number and morphology (*Figure 5A,B*). Peripheral insulin signaling is also reduced in *dHNF4* mutants relative to controls, consistent with their reduced GSIS (*Figure 5C*). This defect in GSIS is due to a tissue-specific requirement for *dHNF4* in the IPCs since IPC-specific RNAi for *dHNF4* resulted in impaired DILP2 secretion into the hemolymph, along with reduced peripheral insulin signaling (*Figure 5D,E* and *Figure 5—figure supplement 1*) (*Park et al., 2014*). Taken together, these data demonstrate that impaired GSIS plays a central role in the diabetic phenotype of *dHNF4* mutants.

## *dHNF4* is required to establish the metabolic state of adult *Drosophila*

As we reported in our prior study of *dHNF4*, the receptor is not expressed in the larval IPCs (*Figure 6A*) (*Palanker et al., 2009*). It is, however, expressed in the IPCs of the adult fly, consistent with its central roles at this stage in GSIS, insulin signaling, and glucose homeostasis (*Figure 6B*). Interestingly, this cell-type specific switch in *dHNF4* expression correlates with a developmental change in IPC physiology. Unlike mammalian β-cells, larval IPCs fail to secrete DILPs in response to dietary glucose (*Geminard et al., 2009*). Adult IPCs, however, display calcium influx, membrane depolarization, and DILP2 secretion in response to glucose, analogous to β-cells (*Alfa et al., 2015*; *Fridell et al., 2009*; *Kreneisz et al., 2010*; *Park et al., 2014*). Along with the temporal induction of dHNF4 expression in adult IPCs, these results suggest that there is a developmental switch in the response to glucose at the onset of adulthood. Consistent with this, glucose feeding activates insulin signaling in adult flies, but not in larvae (*Figure 6C*). This correlates with a ∼ten-fold increase in the basal circulating levels of glucose in adults compared to larvae, which first becomes apparent during the final stages of pupal development (*Figure 6D*) (*Tennessen et al., 2014a*). Moreover, *dHNF4* mutants maintain euglycemia on a normal diet during larval and early pupal stages, but display hyperglycemia just prior to eclosion (*Figure 6D*). Taken together, these observations point to a switch in IPC physiology and glucose homeostasis as *Drosophila* transition into maturity.

The induction of *dHNF4* in the adult IPCs and the adult onset of hyperglycemia in *dHNF4* mutants raise the interesting possibility that this receptor may play a role in coordinating the metabolic

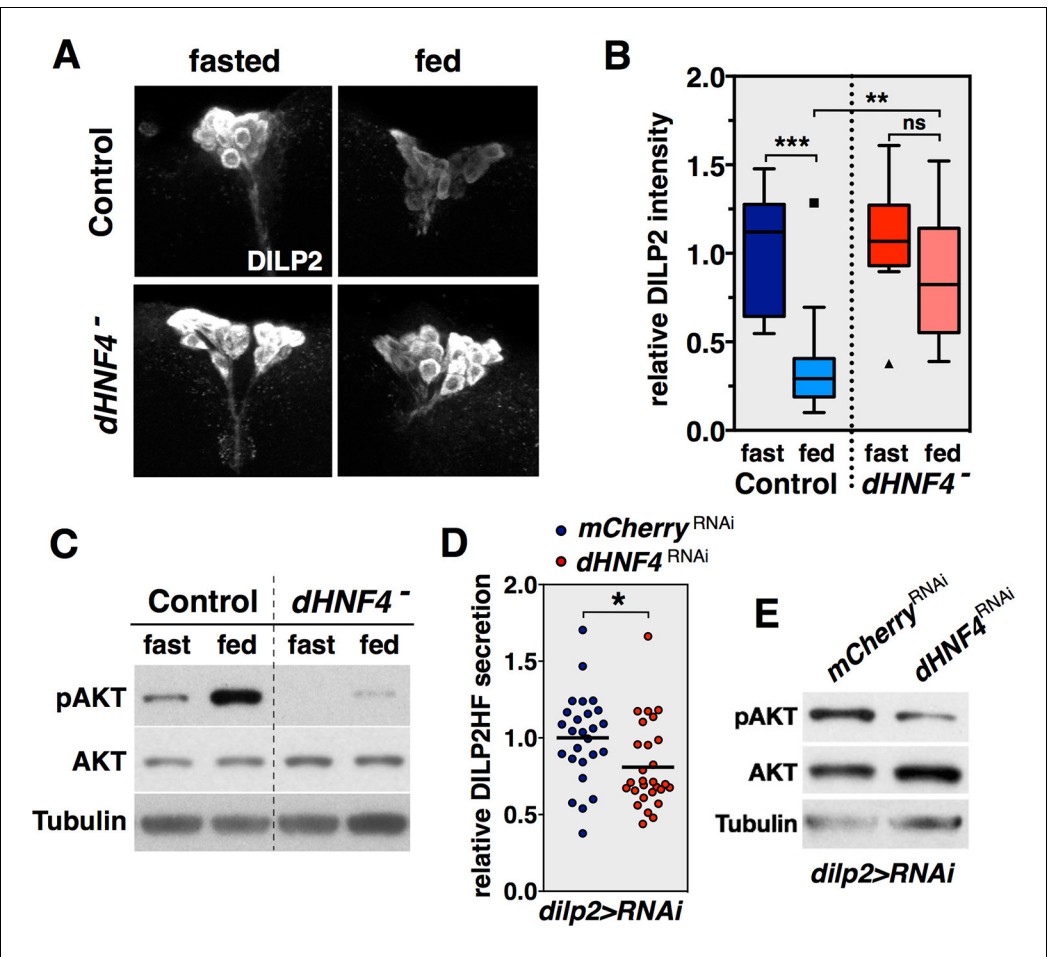

**Figure 5.** *dHNF4* is required for glucose-stimulated DILP2 secretion by the insulin-producing cells. (**A**) Whole-mount staining for DILP2 peptide in brains dissected from adult control and *dHNF4* mutants that were either fasted overnight or re-fed glucose for two hours. (**B**) Quantification of relative DILP2 fluorescent intensity in the IPCs of fasted and glucose-fed controls and *dHNF4* mutants. Data is plotted as a Tukey boxplot with outliers denoted as individual data points (n= 11 ± 3 brains per-condition). Results were consistent between three independent experiments. (**C**) Western blot analysis to detect phosphorylated AKT (pAKT), total AKT, and Tubulin in extracts from controls and *dHNF4* mutants that were either fasted overnight or re-fed glucose for two hours. (**D**) Levels of circulating HA-FLAG-tagged DILP2 (DILP2HF) were assayed in animals with IPC-specific RNAi (TRiP) against either *mCherry* as a control (blue) or *dHNF4* (red) using the *dilp2-GAL4* driver (*dilp2>RNAi*). Data is combined from five independent experiments, each containing 5–6 biological replicates per genotype. The horizontal lines depict the mean value. (**E**) Western blot analysis to detect phosphorylated AKT (pAKT), total AKT, and Tubulin in extracts from *ad libitum* fed adult males with IPC-specific RNAi against either *mCherry* as a control or *dHNF4* using the *dilp2-GAL4* driver (*dilp2>RNAi*). ***$p \leq 0.001$, **$p \leq 0.01$, *$p \leq 0.05$.

The following figure supplement is available for figure 5:

**Figure supplement 1.** *dHNF4* RNAi in the IPCs causes reduced levels of circulating DILP2-HF.

switch toward GSIS and OXPHOS at this stage. Indeed, northern blot analysis of RNA samples isolated from staged wild-type larvae, pupae, and young adults, demonstrate that *dHNF4* expression increases dramatically at the onset of adulthood (*Figure 6E*). Moreover, this temporal pattern of expression is accompanied by increased expression of both nuclear and mitochondrial-encoded *dHNF4* target genes that contribute to OXPHOS as well as *Hex-C*. The expression of these target genes is also reduced in staged *dHNF4* mutants, consistent with our earlier findings that their maximal expression depends on receptor function (*Figure 6E*). Taken together, our results support the

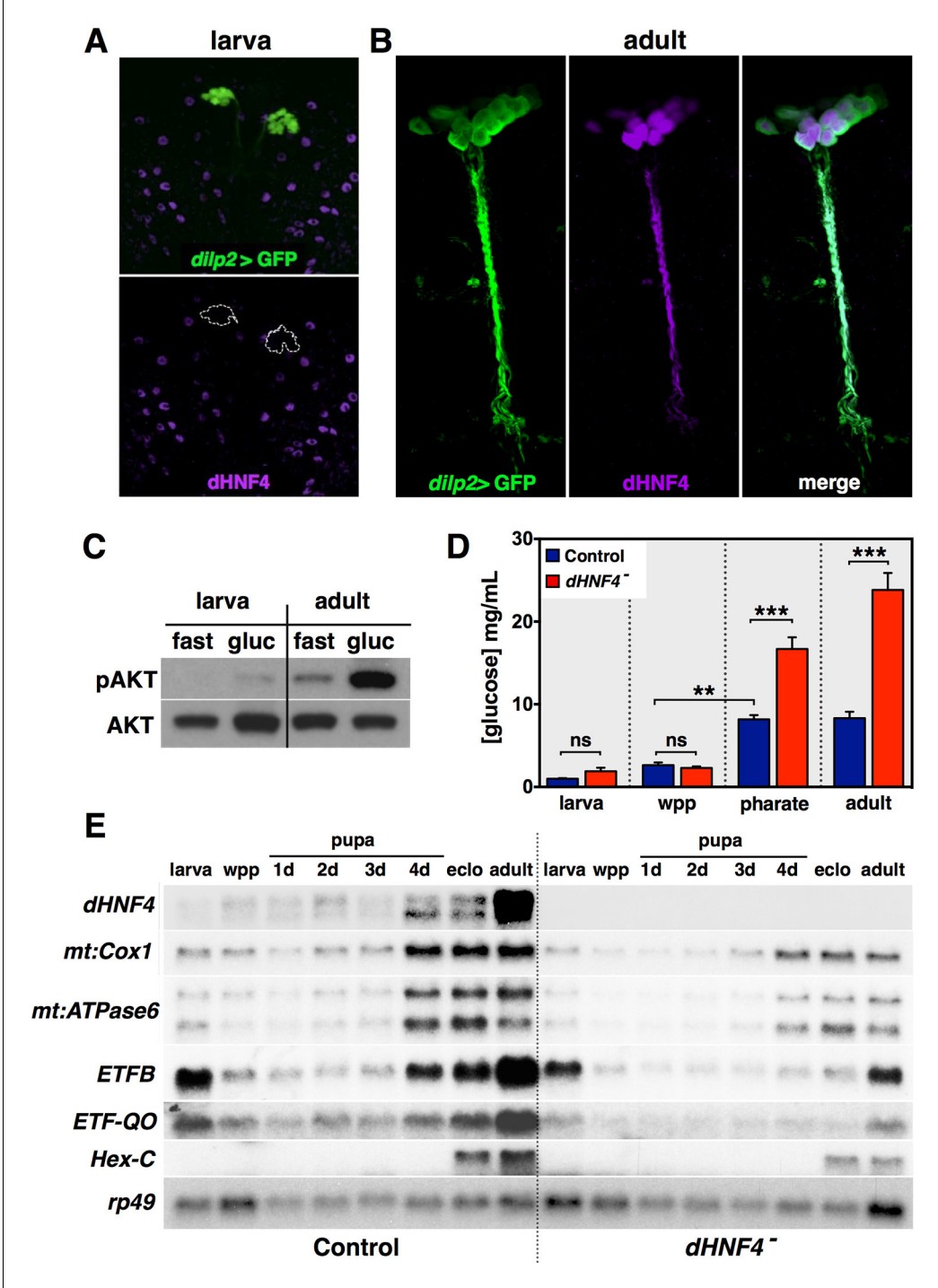

**Figure 6.** *dHNF4* supports a developmental transition toward GSIS and OXPHOS in adult *Drosophila*. (A–B) Whole-mount immunostaining of larval (A) or adult (B) brains to detect dHNF4 protein (magenta) or GFP, which marks the IPCs (*dilp2>GFP*, green). (C) Western blot analysis to detect phosphorylated AKT (pAKT) and total AKT in extracts from $w^{1118}$ third-instar larvae or mature adults that were fasted overnight and re-fed 10% glucose for two hours. (D) Relative levels of free glucose in controls and *dHNF4* mutants staged as either feeding third-instar larvae (larva), white prepupae (wpp), pharate adults (~4 day-old pupae), or mature adults. Data is plotted as the mean ± SEM. ***p≤0.001, **p≤0.01, *p≤0.05. (E) Northern blot analysis of RNA extracted from feeding third-instar larvae (larva), white prepupae (wpp), pupae at one-day intervals, mid-eclosion (eclo), and mature adults. *rp49* is included as a control for loading and transfer.

model that dHNF4 contributes to a metabolic switch in glucose homeostasis at the onset of adulthood that promotes GSIS and OXPHOS to meet the energy demands of the adult fly.

## Discussion

The association of MODY subtypes with mutations in specific genes provides a framework for understanding the monogenic heritability of this disorder as well as the roles of the corresponding pathways in systemic glucose homeostasis. In this paper, we investigate the long-known association between *HNF4A* mutations and MODY1 by characterizing a whole-animal mutant that recapitulates the key symptoms associated with this disorder. We show that *Drosophila* HNF4 is required for both GSIS and glucose clearance in adults, acting in distinct tissues and multiple pathways to maintain glucose homeostasis. We also provide evidence that dHNF4 promotes mitochondrial OXPHOS by regulating nuclear and mitochondrial gene expression. Finally, we show that the expression of *dHNF4* and its target genes is dramatically induced at the onset of adulthood, contributing to a developmental switch toward GSIS and oxidative metabolism at this stage in development. These results provide insights into the molecular basis of MODY1, expand our understanding of the close coupling between development and metabolism, and establish the adult stage of *Drosophila* as an accurate context for genetic studies of GSIS, glucose clearance, and diabetes.

### dHNF4 acts through multiple pathways to regulate glucose homeostasis

*Drosophila HNF4* mutants display late-onset hyperglycemia accompanied by sensitivity to dietary carbohydrates, glucose intolerance, and defects in GSIS – hallmarks of MODY1. These defects arise from roles for *dHNF4* in multiple tissues, including a requirement in the IPCs for GSIS and a role in the fat body for glucose clearance. The regulation of GSIS by *dHNF4* is consistent with the long-known central contribution of pancreatic β-cells to the pathophysiology of MODY1 (*Fajans and Bell, 2011*). Similarly, several MODY-associated genes, including *GCK*, *HNF1A* and *HNF1B*, are important for maintaining normal hepatic function. These distinct tissue-specific contributions to glycemic control may explain why single-tissue *Hnf4A* mutants in mice do not fully recapitulate MODY1 phenotypes and predict that a combined deficiency for the receptor in both the liver and pancreatic β-cells of adults would produce a more accurate model of this disorder.

We used metabolomics, RNA-seq, and ChIP-seq to provide initial insights into the molecular mechanisms by which dHNF4 exerts its effects on systemic metabolism. These studies revealed several downstream pathways, each of which is associated with maintaining homeostasis and, when disrupted, can contribute to diabetes. These include genes identified in our previous study of *dHNF4* in larvae that act in lipid metabolism and fatty acid β-oxidation, analogous to the role of *Hnf4A* in the mouse liver to maintain normal levels of stored and circulating lipids (*Hayhurst et al., 2001*; *Palanker et al., 2009*). Extensive studies have linked defects in lipid metabolism with impaired β-cell function and peripheral glucose uptake and clearance, suggesting that these pathways contribute to the diabetic phenotypes of *dHNF4* mutants (*Prentki et al., 2013*; *Qatanani and Lazar, 2007*). An example of this is *pudgy*, which is expressed at reduced levels in *dHNF4* mutants and encodes an acyl-CoA synthetase that is required for fatty acid oxidation (*Figure 3A*) (*Xu et al., 2012*). Interestingly, *pudgy* mutants have elevated triglycerides, reduced glycogen, and increased circulating sugars, similar to *dHNF4* mutants, suggesting that this gene is a critical downstream target of the receptor. It is important to note, however, that our metabolomic, RNA-seq, and ChIP-seq studies were conducted on extracts from whole animals rather than individual tissues. As a result, some of our findings may reflect compensatory responses between tissues, and some tissue-specific changes in gene expression or metabolite levels may not be detected by our approach. Further studies using samples from dissected tissues would likely provide a more complete understanding of the mechanisms by which *dHNF4* maintains systemic physiology.

Notably, the *Drosophila* GCK homolog encoded by *Hex-C* is expressed at reduced levels in *dHNF4* mutants (*Figure 3A*). The central role of GCK in glucose sensing by pancreatic β-cells as well as glucose clearance by the liver places it as an important regulator of systemic glycemic control. Our functional data supports these associations by showing that *Hex-C* is required in the fat body for proper circulating glucose levels, analogous to the role of GCK in mammalian liver (*Figure 4B*) (*Postic et al., 1999*). Unlike mice lacking *GCK* in the β-cells, however, we do not see an effect on

glucose homeostasis when *Hex-C* is targeted by RNAi in the IPCs. This is possibly due to the presence of a second *GCK* homolog in *Drosophila,* Hex-A, which could act alone or redundantly with Hex-C to mediate glucose sensing by the IPCs. In mammals, *GCK* expression is differentially regulated between hepatocytes and β-cells through the use of two distinct promoters, and studies in rats have demonstrated a direct role for HNF4A in promoting *GCK* expression in the liver (*Roth et al., 2002*). Our findings suggest that this relationship has been conserved through evolution. In addition, the association between *GCK* mutations and MODY2 raise the interesting possibility that defects in liver GCK activity may contribute to the pathophysiology of both MODY1 and MODY2.

Interestingly, gene ontology analysis indicates that the up-regulated genes in *dHNF4* mutants correspond to the innate immune response pathways in *Drosophila* (*Supplementary file 2*). This response parallels that seen in mice lacking *Hnf4A* function in enterocytes, which display intestinal inflammation accompanied by increased sensitivity to DSS-induced colitis and increased permeability of the intestinal epithelium, similar to humans with inflammatory bowel disease (*Ahn et al., 2008*; *Babeu and Boudreau, 2014*; *Cattin et al., 2009*). Disruption of *Hnf4A* expression in Caco-2 cells using shRNA resulted in changes in the expression of genes that act in oxidative stress responses, detoxification pathways, and inflammatory responses, similar to the effect we see in *dHNF4* mutants (*Marcil et al., 2010*). Moreover, mutations in human *HNF4A* are associated with chronic intestinal inflammation, irritable bowel disease, ulcerative colitis, and Crohn's disease, suggesting that these functions are conserved through evolution (*UK IBD Genetics Consortium et al., 2009*; *Marcil et al., 2012*; *van Sommeren et al., 2011*). Taken together, these results support the hypothesis that *dHNF4* plays an important role in suppressing an inflammatory response in the intestine. Future studies are required to test this hypothesis in *Drosophila*. In addition, further work is required to better define the regulatory functions of HNF4 that are shared between *Drosophila* and mammals. Although our work described here suggests that key activities for this receptor have been conserved in flies and mammals, corresponding to the roles of HNF4 in the IPCs (β-cells) for GSIS, fat body (liver) for lipid metabolism and glucose clearance, and intestine to suppress inflammation, there are likely to be divergent roles as well. One example of this is the embryonic lethality of *Hnf4A* mutant mice, which is clearly distinct from the early adult lethality reported here for *dHNF4* mutants. Further studies are required to dissect the degree to which the regulatory functions of this receptor have been conserved through evolution.

It is also important to note that mammalian *Hnf4A* plays a role in hepatocyte differentiation and proliferation in addition to its roles in metabolism (*Bonzo et al., 2012*; *Li et al., 2000*). This raises the possibility that early developmental roles for *dHNF4* could impact the phenotypes we report here in adults. Indeed, all of our studies involve zygotic *dHNF4* null mutants that lack function throughout development. In an effort to address this possibility and distinguish developmental from adult-specific functions, we are constructing a conditional *dHNF4* mutant allele using CRISPR/Cas9 technology. Future studies using this mutation should allow us to conduct a detailed phenotypic analysis of this receptor at different stages of *Drosophila* development.

It is also interesting to speculate that our functional studies of *dHNF4* uncover more widespread roles for MODY-associated genes in glycemic control, in addition to the link with MODY2 described above. *HNF1A* and *HNF1B*, which are associated with MODY3 and MODY5, respectively, act together with *HNF4A* in an autoregulatory circuit in an overlapping set of tissues, with HNF4A proposed to be the most upstream regulator of this circuit (*Boj et al., 2001*; *Nagaki and Moriwaki, 2008*). The observation that *Drosophila* do not have identifiable homologs for *HNF1A* and *HNF1B* raises the interesting possibility that *dHNF4* alone replaces this autoregulatory circuit in more primitive organisms. The related phenotype of these disorders is further emphasized by cases of MODY3 that are caused by mutation of an HNF4A binding site within the *HNF1A* promoter (*Gragnoli et al., 1997*). Consistent with this link, MODY1, MODY3 and MODY5 display similar features of disease complication and progression, and studies of HNF1A and HNF4A in INS-1 cells have implicated roles for these transcription factors in promoting mitochondrial metabolism in β-cells (*Wang et al., 2002*). In line with this, mitochondrial diabetes is clearly age progressive, as are MODY1, 3, and 5, but not MODY2, which represents a more mild form of this disorder. Furthermore, the severity and progression of MODY3 is significantly enhanced when patients carry an additional mutation in either HNF4A or mtDNA (*Forlani et al., 2010*). Overall, these observations are consistent with the well-established multifactorial nature of diabetes, with multiple distinct metabolic insults contributing to disease onset.

## dHNF4 regulates nuclear and mitochondrial gene expression

Our RNA-seq analysis supports a role for *dHNF4* in coordinating mitochondrial and nuclear gene expression (*Supplementary file 1* and *Figure 3A*). This is represented by the reduced expression of transcripts encoded by the mitochondrial genome, along with effects on nuclear-encoded genes that act in mitochondria. In addition, ChIP-seq revealed that several of the nuclear-encoded genes are direct targets of the receptor. Mitochondrial defects have well-established links to diabetes-onset, with mutations in mtDNA causing maternally-inherited diabetes and mitochondrial OXPHOS playing a central role in both GSIS and peripheral glucose clearance (*Sivitz and Yorek, 2010*). Consistent with this, our functional studies indicate that dHNF4 is required to maintain normal mitochondrial function and that defects in this process contribute to the diabetic phenotypes in *dHNF4* mutants.

It is important to note that the number of direct targets for dHNF4 in the nucleus is difficult to predict with our current dataset. A relatively low signal-to-noise ratio in our ChIP-seq experiment allowed us to identify only 37 nuclear-encoded genes as high confidence targets by fitting the criteria of proximal dHNF4 binding along with reduced expression in *dHNF4* mutants (*Figure 3B* and *Supplementary file 3*). Future ChIP-seq studies will allow us to expand this dataset to gain a more comprehensive understanding of the scope of the dHNF4 regulatory circuit and may also reveal tissue-restricted targets that are more difficult to detect. Nonetheless, almost all of the genes identified as direct targets for dHNF4 regulation correspond to genes involved in mitochondrial metabolism, including the TCA cycle, OXPHOS, and lipid catabolism, demonstrating that this receptor has a direct impact on these critical downstream pathways that influence glucose homeostasis (*Figure 3—figure supplement 3*, *Supplementary file 3*).

## dHNF4 is required for mitochondrial function

An unexpected and significant discovery in our studies is that dHNF4 is required for mitochondrial gene expression and function. Several lines of evidence support the model that dHNF4 exerts this effect through direct regulation of mitochondrial transcription, although a number of additional experiments are required to draw firm conclusions on this regulatory connection. First, most of the 13 protein-coding genes in mtDNA are underexpressed in *dHNF4* mutants (*Figure 3A*, *Supplementary file 1*). Our lab and others have conducted RNA-seq studies of *Drosophila* nuclear transcription factor mutants and, to our knowledge, similar effects on mitochondrial gene expression have not been reported previously. Second, dHNF4 protein is abundantly bound to the control region of the mitochondrial genome, representing the fifth strongest enrichment peak in our ChIP-seq dataset (*Figure 3C*). Although the promoters in *Drosophila* mtDNA have not yet been identified, the site bound by dHNF4 corresponds to a predicted promoter region for *Drosophila* mitochondrial transcription and coincides with the location of the major divergent promoters in human mtDNA (*Garesse and Kaguni, 2005*; *Roberti et al., 2006*). It is unlikely that the abundance of mtDNA relative to nuclear DNA had an effect on our ChIP-seq peak calling because the MACS2 platform used for our analysis accounts for local differences in read depth across the genome (including the abundance of mtDNA). In addition, although the D-loop in mtDNA has been proposed to contribute to possible false-positive ChIP-seq peaks in mammalian studies (*Marinov et al., 2014*), the D-loop structure is not present in *Drosophila* mtDNA (*Rubenstein et al., 1977*). Nonetheless, additional experiments are required before we can conclude that this apparent binding is of regulatory significance for mitochondrial function. Third, the effects on mitochondrial gene expression do not appear to be due to reduced mitochondrial number in *dHNF4* mutants (*Figure 3—figure supplement 2A*). This is consistent with the normal expression of *mt:Cyt-b* in *dHNF4* mutants (*Figure 3A*), which has a predicted upstream promoter that drives expression of the *mt:Cyt-b* and mt:*ND6* operon (although we could not detect *mt:ND6* RNA in our northern blot studies) (*Berthier et al., 1986*; *Roberti et al., 2006*). Fourth, immunostaining for dHNF4 shows cytoplasmic protein that overlaps with the mitochondrial marker ATP5A, in addition to its expected nuclear localization (*Figure 3—figure supplement 2B*). Some of the cytoplasmic staining, however, clearly fails to overlap with the mitochondrial marker, making it difficult to draw firm conclusions from this experiment. Multiple efforts to expand on this question biochemically with subcellular fractionation studies have been complicated by abundant background proteins that co-migrate with the receptor in mitochondrial extracts. We are currently developing new reagents to detect the relatively low levels of endogenous dHNF4 protein in

mitochondria, including use of the CRISPR/Cas9 system for the addition of specific epitope tags to the endogenous *dHNF4* locus. Finally, we observe multiple hallmarks of mitochondrial dysfunction, including elevated pyruvate and lactate, specific alterations in TCA cycle metabolites, reduced mitochondrial membrane potential, reduced levels of ATP, and fragmented mitochondrial morphology. These phenotypes are consistent with the reduced expression of key genes involved in mitochondrial OXPHOS (*Figure 3A* and *Supplementary file 1*), and studies showing that decreased mitochondrial membrane potential and ATP production are commonly associated with mitochondrial fragmentation (*Mishra and Chan, 2014*; *Toyama et al., 2016*).

Although unexpected, our proposal that dHNF4 may directly regulate mitochondrial gene expression is not unprecedented. A number of nuclear transcription factors have been localized to mitochondria, including ATFS-1, MEF2D, CREB, p53, STAT3, along with several nuclear receptors, including the estrogen receptor, glucocorticoid receptor, and the p43 isoform of the thyroid hormone receptor (*Leigh-Brown et al., 2010*; *Nargund et al., 2015*; *Szczepanek et al., 2012*). The significance of these observations, however, remains largely unclear, with few studies demonstrating regulatory functions within mitochondria. In addition, these factors lack a canonical mitochondrial localization signal at their amino-terminus, leaving it unclear how they achieve their subcellular distribution (*Marinov et al., 2014*). In contrast, one of the five mRNA isoforms encoded by *dHNF4*, *dHNF4-B*, encodes a predicted mitochondrial localization signal in its 5′-specific exon, providing a molecular mechanism to explain the targeting of this nuclear receptor to this organelle. Efforts are currently underway to conduct a detailed functional analysis of *dHNF4-B* by using the CRISPR/Cas9 system to delete its unique 5′ exon, as well as establishing transgenic lines that express a tagged version of *dHNF4-B* under UAS control. Future studies using these reagents, along with our *dHNF4* mutants, should allow us to dissect the nuclear and mitochondrial functions of this nuclear receptor and their respective contributions to systemic physiology.

Finally, it is interesting to speculate whether the role for dHNF4 in mitochondria is conserved in mammals. A few papers have described the regulation of nuclear-encoded mitochondrial genes by HNF4A (*Rodriguez et al., 1998*; *Wang et al., 2000*). In addition, several studies have detected cytoplasmic *Hnf4A* by immunohistochemistry in tissue sections, including in postnatal pancreatic islets (*Miura et al., 2006*; *Nammo et al., 2008*) and hepatocytes (*Bell and Michalopoulos, 2006*; *Soutoglou et al., 2000*; *Sun et al., 2007*; *Yanger et al., 2013*). Moreover, the regulation of nuclear/cytoplasmic shuttling of HNF4A has been studied in cultured cells (*Soutoglou et al., 2000*). The evolutionary conservation of the physiological functions of HNF4A, from flies to mammals, combined with these prior studies, argue that more effort should be directed at defining the subcellular distribution of HNF4A protein and its potential roles within mitochondria. Taken together with our studies in *Drosophila*, this work could provide new directions for understanding HNF4 function and MODY1.

## dHNF4 contributes to an adult switch in metabolic state

Physiological studies by George Newport in 1836 noted that holometabolous insects reduce their respiration during metamorphosis leading to a characteristic "U-shaped curve" in oxygen consumption (*Needham, 1929*; *Newport, 1836*). Subsequent classical experiments in *Lepidoptera*, *Bombyx*, *Rhodnius* and *Calliphora* showed that this reduction in mitochondrial respiration during metamorphosis and dramatic rise in early adults is seen in multiple insect species, including *Drosophila* (*Bodine, 1925*; *Merkey et al., 2011*). Consistent with this, the activity of oxidative enzyme systems and the levels of ATP also follow a "U-shaped curve" during development as the animal transitions from a non-feeding pupa to a motile and reproductively active adult fly (*Agrell, 1953*). Although first described over 150 years ago, the regulation of this developmental increase in mitochondrial activity has remained undefined. Here we show that this temporal switch is dependent, at least in part, on the dHNF4 nuclear receptor. The levels of *dHNF4* expression increase dramatically at the onset of adulthood, accompanied by the expression of downstream genes that act in glucose homeostasis and mitochondrial OXPHOS (*Figure 6E*). This coordinate transcriptional switch is reduced in *dHNF4* mutants, indicating that the receptor plays a key role in this transition. Importantly, the timing of this program correlates with the onset of *dHNF4* mutant phenotypes in young adults, including sugar-dependent lethality, hyperglycemia, and defects in GSIS, indicating that the upregulation of *dHNF4* expression in adults is of functional significance. It should also be noted, however, that *dHNF4* target genes are still induced at the onset of adulthood in *dHNF4* mutants, albeit at lower levels, indicating that other regulators contribute to this switch in metabolic state (*Figure 6E*). Nonetheless, the

timing of the induction of *dHNF4* and its target genes in early adults, and its role in promoting OXPHOS, suggest that this receptor contributes to the end of the "U-shaped curve" and directs a systemic transcriptional switch that establishes an optimized metabolic state to support the energetic demands of adult life.

Interestingly, a similar metabolic transition towards OXPHOS was recently described in *Drosophila* neuroblast differentiation, mediated by another nuclear receptor, EcR (*Homem et al., 2014*). Although this occurs during early stages of pupal development, prior to the *dHNF4*-mediated transition at the onset of adulthood, the genes involved in this switch show a high degree of overlap with *dHNF4* target genes that act in mitochondria, including *ETFB*, components of Complex IV, *pyruvate carboxylase*, and members of the α-ketoglutarate dehydrogenase complex. This raises the possibility that *dHNF4* may contribute to this change in neuroblast metabolic state and play a more general role in supporting tissue differentiation by promoting OXPHOS.

To our knowledge, only one other developmentally coordinated switch in systemic metabolic state has been reported in *Drosophila* and, intriguingly, it is also regulated by a nuclear receptor. *Drosophila* Estrogen-Related Receptor (dERR) acts in mid-embryogenesis to directly induce genes that function in biosynthetic pathways related to the Warburg effect, by which cancer cells use glucose to support rapid proliferation (*Tennessen et al., 2011*; *Tennessen et al., 2014b*). This switch toward aerobic glycolysis favors lactate production and flux through biosynthetic pathways over mitochondrial OXPHOS, supporting the ~200-fold increase in mass that occurs during larval development. Taken together with our work on dHNF4, these studies define a role for nuclear receptors in directing temporal switches in metabolic state that meet the changing physiological needs of different stages in development. Further studies should allow us to better define these regulatory pathways as well as determine how broadly nuclear receptors exert this role in coupling developmental progression with systemic metabolism.

Although little is known about the links between development and metabolism, it is likely that coordinated switches in metabolic state are not unique to *Drosophila*, but rather occur in all higher organisms in order to meet the distinct metabolic needs of an animal as it progresses through its life cycle. Indeed, a developmental switch towards OXPHOS in coordination with the cessation of growth and differentiation appears to be a conserved feature of animal development. Moreover, as has been shown for cardiac hypertrophy, a failure to coordinate metabolic state with developmental context can have an important influence on human disease (*Lehman and Kelly, 2002*).

## Adult *Drosophila* as a context for genetic studies of GSIS and diabete

In addition to promoting a transition toward systemic oxidative metabolism in adult flies, *dHNF4* also contributes to a switch in IPC physiology that supports GSIS. dHNF4 is not expressed in larval IPCs, but is specifically induced in these cells at adulthood (*Figure 6A,B*). Similarly, the fly homologs of the mammalian ATP-sensitive potassium channel subunits, Sur1 and Kir6, which link OXPHOS and ATP production to GSIS, are not expressed in the larval IPCs but are expressed during the adult stage (*Fridell et al., 2009*; *Kim and Rulifson, 2004*). They also appear to be active at this stage as cultured IPCs from adult flies undergo calcium influx and membrane depolarization upon exposure to glucose or the anti-diabetic sulfonylurea drug glibenclamide (*Kreneisz et al., 2010*). In addition, reduction of the mitochondrial membrane potential in adult IPCs by ectopic expression of an uncoupling protein is sufficient to reduce IPC calcium influx, elevate whole-animal glucose levels, and reduce peripheral insulin signaling (*Fridell et al., 2009*). This switch in IPC physiology is paralleled by a change in the nutritional signals that trigger DILP release. Amino acids, and not glucose, stimulate DILP2 secretion by larval IPCs (*Geminard et al., 2009*). Rather, glucose is sensed by the corpora cardiaca in larvae, a distinct organ that secretes adipokinetic hormone, which acts like glucagon to maintain carbohydrate homeostasis during larval stages (*Kim and Rulifson, 2004*; *Lee and Park, 2004*). Interestingly, this can have an indirect effect on the larval IPCs, triggering DILP3 secretion in response to dietary carbohydrates (*Kim and Neufeld, 2015*). Adult IPCs, however, are responsive to glucose for DILP2 release (*Park et al., 2014*) (*Figure 5A,D*). In addition, *dHNF4* mutants on a normal diet maintain euglycemia during larval and early pupal stages, but display hyperglycemia at the onset of adulthood, paralleling their lethal phase on a normal diet (*Figure 6D*). Taken together, these observations support the model that the IPCs change their physiological state during the larval-to-adult transition and that *dHNF4* contributes to this transition toward GSIS. The observation that glucose is a major circulating sugar in adults, but not larvae, combined with its ability to

stimulate DILP2 secretion from adult IPCs, establishes this stage as an experimental context for genetic studies of glucose homeostasis, GSIS, and diabetes. Functional characterization of these pathways in adult *Drosophila* will allow us to harness the power of model organism genetics to better understand the regulation of glucose homeostasis and the factors that contribute to diabetes.

## Materials and methods

### *Drosophila* strains and media

All genetic studies used a transheterozygous combination of *dHNF4* null alleles (*dHNF4$^{\Delta 17}$*/*dHNF4$^{\Delta 33}$*) and genetically-matched controls that were transheterozygous for precise excisions of the EP2449 and KG08976 P-elements, as described previously (*Palanker et al., 2009*). Sugar diets were made using 8% yeast, 1% agar, 0.05% MgSO$_4$, 0.05% CaCl$_2$, and either 3%, 9% or 15% dietary sugar (2:1 ratio of glucose to sucrose, percentages represent weight/final food volume. 10 ml/L tegosept and 6 ml/L propionic acid were added just prior to pouring). Fasting was achieved by using 1% agar as a medium. For adult studies, 8–10 day old males were selected for all studies unless otherwise indicated.

### GAL4/UAS lines for tissue-specific RNAi studies

The following GAL4 driver lines were used for tissue-specific expression experiments: Fat body: *r4-GAL4* (*Lee and Park, 2004*), midgut: *mex-GAL4* (*Phillips and Thomas, 2006*), IPCs: *yw; UAS-Dicer2; dilp2$^1$ dilp2HF dilp2-GAL4* (*Park et al., 2014*). RNAi lines used in this study include: *UAS-dHNF4$^{RNAi}$* (TRiP 29375 used in *Figure 4A*, *Figure 4—figure supplement 1A*, *Figure 5D,E*; VDRC 12692 used in *Figure 4—figure supplement 1A–B*, *Figure 5—figure supplement 1*), *UAS-mCherry$^{RNAi}$* (TRiP 35785), *UAS-Cox5a$^{RNAi}$* (TRiP 27548), *UAS-CIA30$^{RNAi}$* (TRiP 55660), *UAS-ATPsynβ$^{RNAi}$* (TRiP 28056), *UAS-Scs-α$^{RNAi}$* (TRiP 51807), *UAS-CG5599$^{RNAi}$* (TRiP 32876), and *UAS-HexC$^{RNAi}$* (TRiP 57404 used in *Figure 4B–D*, and VDRC 35337 and VDRC 35338 used in *Figure 4—figure supplement 3*).

### Lifespan studies

The *dHNF4* mutant lethal phase was assessed by raising 50–60 newly hatched first instar larvae in vials of standard laboratory media at 25°C and scoring for survival through embryonic hatching, wandering third instar, puparium formation, eclosion, and survival through the first day of adulthood (*Figure 1A*). Eclosion rates were scored in a similar manner for *dHNF4* mutants and genetically matched controls raised on the 3%, 9% or 15% sugar diets. For adult survival studies on different diets (*Figure 1C*), 50–60 newly hatched first instar larvae were placed in vials with the 3% sugar diet and raised until five days of adulthood. These mature adults were then transferred to fresh vials of 3%, 9%, or 15% dietary sugar with approximately 25 males and 25 females per vial at 25°C and scored daily for lethality. Flies were transferred to fresh vials every 2–3 days. At least three replicate vials were analyzed per condition and each experiment was repeated at least three times with similar results.

### Developmental staging

For the developmental time course northern blot, we collected 0–12 hr feeding third instar larvae, pupae at 24, 48 or 72 hr after puparium formation, stage P12 males (newly formed pharate adult with visible sex combs) for 4 day pupae, and males at mid-eclosion from the pupal case (to ensure that all *dHNF4* mutants were alive) (n=15 animals per sample). Animals collected for developmental RNA and glycemia measurements (*Figure 6D–E*) were raised on the 15% sugar diet, except for the adults, which were reared on the 3% sugar diet and transferred to the 15% sugar diet after eclosion.

### Metabolic assays

Glycogen, trehalose, and free glucose levels were determined using the Hexokinase (HK) and/or Glucose Oxidase (GO) assay kits (Sigma GAHK20, GAGO20) as previously described, with approximately six biological replicates (n=5 animals per sample) assayed per condition (*Tennessen et al., 2014a*). Total protein levels were determined in parallel by Bradford assay to control for potential variations in sample homogenization and/or animal size. Hemolymph glucose measurements were determined by centrifuging 60–100 adult males in a prepared zymogen barrier column (Zymo

Research Corporation C1006) at 9000 g for 5 min at 4°C, as described (*Park et al., 2014*). The hemolymph flow-though was diluted 1:100 in 1xPBS prior to heat treatment, followed by analysis using the HK assay kit.

## ATP assay

Mature adult males were placed on 10% sucrose + 1% agar overnight prior to analysis. Six biological replicates per genotype (n=5 males per sample) were analyzed for ATP levels by using a luminescence assay kit (Molecular Probes ATP kit; A22066, ) as described (*Tennessen et al., 2014a*).

## GC/MS Metabolomic analysis

Animals were raised on the 3% sugar diet until five days of adulthood and transferred to a fresh vial with either the 3% or 15% sugar diet for three days prior to collection. Twenty adult males per sample were snap-frozen in liquid nitrogen and prepared for analysis by gas chromatography-mass spectrometry (GC/MS) as described (*Tennessen et al., 2014a*). Data is presented from three independent experiments, each consisting of 5–6 biological replicates per condition. Sample preparation and GC/MS analysis were performed by the Metabolomics Core Research Facility at the University of Utah School of Medicine.

## Immunostaining

Tissues were dissected in cold 1xPBS, fixed in 4% paraformaldehyde for 20-30 min at room temperature, and washed once quickly followed by three washes for 20 min each in PBS, 0.2% Triton-X100. Samples were blocked using 5% normal goat serum for 1–2 hr at room temperature and incubated with primary antibodies for a minimum of 24 hr at 4°C, washed, and incubated with secondary antibody at 4°C for 24 hr. Images were acquired using an Olympus FV1000 confocal microscope and analyzed by Volocity software to generate Z-stack projections and overlay images. The following antibodies were used for immunofluorescence studies: rat anti-dHNF4 (*Palanker et al., 2009*), rat anti-DILP2 (a gift from P. Leopold), rabbit anti-GFP (Molecular Probes A-6455), chicken anti-GFP (Abcam 13970), and mouse anti-ATP5A (Abcam 14748).

## Quantification of DILP2 fluorescence

Dissected brains stained for DILP2 peptide were mounted in SlowFade® Gold (Life Technologies) and imaged using an Olympus FV1000 confocal microscope and 60X water-immersion objective. Scan settings were fixed to be identical for all captured images, with Z-stack limits set to encompass the entire depth of DILP2 fluorescence for each brain. Maximum projection Z-stacks were analyzed using Volocity software to calculate mean pixel intensity of DILP2 fluorescence within the IPCs (selected by ROI tool) for each set of IPCs analyzed.

## Northern blot analysis

Total RNA was extracted from groups of 10–20 animals using TriPure isolation reagent (Roche). RNA was fractionated by 1% formaldehyde gel electrophoresis at a constant voltage (70V) for ~3.5 hr prior to transfer to a nylon membrane overnight. After transfer, RNA was UV cross-linked to the membrane and placed in hybridization buffer (5 mls formamide, 2 mls 10x PIPES buffer, pH 6.5, 2 mls ddH$_2$0, 1 ml 10% SDS, 100 µl sheared herring sperm ssDNA) for two hours at 42°C prior to the addition of radiolabeled probe. Probes were generated by Klenow-mediated random primer amplification of purified template DNA corresponding to approximately 100–500 bp of the transcript of interest, allowing incorporation of $^{32}$P radiolabeled-nucleotide (dCTP, PerkinElmer). Probes were incubated with membranes overnight, and the hybridized membranes were washed and exposed to X-ray film at -80°C using two intensifying screens.

## RNA-seq

*dHNF4* mutants and genetically-matched controls were reared on the 3% sugar diet for five days of adulthood and transferred to the 15% sugar diet for 3 days prior to extracting total RNA. Eight biological replicates per genotype (n=20 males per sample) were collected using TriPure isolation reagent (Roche). Pairs of biological replicates were pooled to obtain four biological replicates for further purification using an RNeasy kit (QIAGEN). RNA quality was subsequently analyzed using an

Agilent Bioanalyzer RNA 6000. PolyA-selected RNAs from each biological replicate were then assembled into barcoded libraries and pooled into a single-flow cell lane for Illumina HighSeq2000 50-cycle single-read sequencing, which produced ≥21.9 million reads per sample. Standard replicate RNA-seq analysis was performed using USeq and DESeq analysis packages with alignment to the *Drosophila melanogaster* dm3 genome assembly. Transcripts meeting a cutoff of ≥1.5 fold differ- ence in mRNA abundance and FDR ≤1% were considered as differentially expressed genes. RNA quality control, library preparation, sequencing, and data analysis were performed at the University of Utah High Throughput Genomics and Bioinformatics Core Facilities. Although *mt:Cyt-b* and *mt: ND6* are included among the down-regulated genes in our RNA-Seq dataset (*Supplementary file 1*), these represent false positives as demonstrated by subsequent repeated northern blot hybridiza- tion studies for *mt:Cyt-b* (*Figure 3A*). We are also unable to detect *mt:ND6* mRNA in either mutant or control flies, consistent with a previous report (*Berthier et al., 1986*).

## ChIP-seq

Chromatin isolation and immunoprecipitation were performed as described (*Schwartz et al., 2003*). $w^{1118}$ flies were reared on standard cornmeal-based lab medium and 1–1.5 g of mature adults were homogenized in ice-cold buffer A (0.3 M sucrose, 2 mM MgOAc, 3 mM $CaCl_2$, 10 mM Tris-Cl [pH 8], 0.3% Triton X-100, 0.5 mM dithiothreitol, 1 Roche protease inhibitor tablet per 10 ml) for 1.5 min using a Brinkmann Homogenizer Polytron PT 10/35. The homogenate was filtered through a layer of 100 μm-pore nylon mesh into a pre-chilled glass-Teflon homogenizer, stroked on ice 35 times using a B pestle, and filtered through two layers of 40-μm-pore nylon mesh prior to adding one volume of warm cross-linking buffer (0.1 M NaCl, 1 mM EDTA, 0.5 mM EGTA, 50 mM Tris [pH 8], pre-warmed to 40°C) to bring the sample to room temperature for crosslinking (0.3% formaldehyde for 3 min). 2.5 M glycine was added to a final concentration of 125 mM to stop the crosslinking after 3 min. The mixture was pelleted for resuspension into 10 ml of RIPA buffer (140 mM NaCl, 10 mM Tris-Cl [pH 8.0], 1 mM EDTA, 1% Triton, 0.3% SDS, 0.1% sodium deoxycholate, and Roche protease inhibi- tor cocktail) for sonication. The sonicated material was centrifuged at 20,000 *g*, and the supernatant was distributed into 500 μl aliquots that were snap frozen in liquid nitrogen. To each aliquot, 1 ml of cold RIPA buffer (without SDS) was added prior to removing 150 μl for an input control, and then 5 μl of polyclonal affinity-purified anti-dHNF4 antibody was added to each sample for immunoprecipi- tation overnight at 4°C (*Palanker et al., 2009*). 50 μl of prepared Protein G Dynabeads (Life Technol- ogies) were added to each sample and incubated for 4 hr at 4°C prior to wash and elution using a magnetic stand. Washes were performed for 3 min each at 4°C with 1 ml of the following ice-cold solutions: three times in low salt wash buffer (0.1% SDS, 1% Triton X-100, 2 mM EDTA, 20 mM Tris- Cl [pH 8.0], 150 mM NaCl), one time in high salt wash buffer (0.1% SDS, 1% Triton X-100, 2 mM EDTA, 20 mM Tris-Cl [pH 8.0], 500 mM NaCl), once in LiCl wash buffer (0.25 M LiCl, 1% NP-40, 1% deoxycholate, 1 mM EDTA, 10 mM Tris-Cl [pH 8.0]), and twice with Tris-EDTA. Protein-DNA com- plexes were eluted in 1.5 ml DNA-low bind tubes (Eppendorf) using two 15-minute washes with 250 μl elution buffer (1% SDS, 0.1 M NaHCO3). NaCl was added to a final concentration of 0.2 M to reverse the crosslinks of IP and input control samples, followed by an overnight incubation at 65°C. dHNF4-bound DNA was purified by using PCR-purification columns (Qiagen), and pooled to acquire four replicates of dHNF4-IP chromatin and input controls. Barcoded libraries for dHNF4-IP and input control samples were generated by the University of Utah High Throughput Genomics core facility and sequenced in a single lane Illumina HiSeq 50-cycle single-read sequencing. Data analysis was performed by the Bioinformatics Core at the University of Utah School of Medicine using USeq Scan- Seqs (FDR80) as well as Model-based Analysis for ChIP-seq 2.0 (MACS2) (*Zhang et al., 2008*) with an FDR cutoff of 1% (FDR20), identifying 68 enrichment regions. Nearest neighboring genes were compiled using USeq FindNeighboringGenes and UCSC dm3 EnsGenes gene tables and were com- pared to our RNA-seq dataset to identify proximal genes that are also misexpressed in *dHNF4* mutants, highlighting these as direct targets of dHNF4.

## Immunoblotting

Samples were homogenized in Laemmli sample buffer with protease and phosphatase inhibitor cock- tails, resolved by SDS-PAGE (10% acrylamide), transferred to PVDF membrane overnight at 4°C, and blocked with 5% BSA prior to immunoblotting. Antibodies used for western blots include rabbit

anti-pAKT (Cell Signaling #4054), rabbit anti-AKT (Cell Signaling #4691), mouse anti-SDHA (Abcam 137756), rabbit anti-SDHB (a generous gift from D. Winge, generated against yeast SDH2/SDHB), mouse anti-ATP5A (Abcam 14748), and mouse anti-βTubulin (Chemicon MAB380).

## DILP2HF ELISA for measuring circulating DILP2

Transgenic lines expressing HA-FLAG tagged DILP2 (DILP2-HF) peptide were used for ELISA assays, essentially as described (*Park et al., 2014*). The indicated *UAS-RNAi* line or *w^1118* controls (*Figure 5D* and *Figure 5—figure supplement 1*) were crossed to *yw; UAS-Dcr2; dilp2^1 DILP2HF dilp2-GAL4* flies. Adult male progeny were either fed *ad libitum* (*Figure 5—figure supplement 1*) or fasted overnight and then re-fed the 15% sugar diet for 45 min prior to analysis (*Figure 5D*). The posterior end of the abdomen was removed using micro-dissection scissors and placed in 60 µl of PBS, using 10 males per biological replicate. Tubes were vortexed for 20 min, after which 50 µl of supernatant was collected in a fresh tube. Samples were homogenized in 600 µl of PBS, 1% Triton X-100. HA-FLAG peptide standards from 0, 20, 40 80, 160, 320 and 640 pg/ml were generated for a linear standard curve. 50 µl of circulating DILP2HF, total DILP2HF, or peptide standards were added to 50 µl of anti-HA-peroxidase (5 ng/ml, Roche 3F10) in PBS (with or without 1% Triton X-100) and then pipetted into a 96-well ELISA plate (Thermo Scientific MaxiSorp Immulon 4 HBX, Cat# 3855) coated with mouse anti-FLAG antibody (Sigma F1804, M2 monoclonal). Samples were incubated at 4°C overnight and washed 6 times with PBS, 0.2% Tween-20. 100 µl of 1-Step Ultra TMB ELISA Substrate (Thermo Scientific 34029) was added to the plate and incubated at room temperature for 30 min. 100 µl of 2M sulfuric acid was added to stop the reaction and the absorbance was measured immediately at 450 nm. Circulating DILP2HF (pg/fly) versus total remaining peptide was calculated to determine the percent secretion relative to controls (n≥4 biological replicates per condition).

## Clonal analysis

Virgin female flies (*yw,hsFLP,UAS-GFP; tub-GAL80,FRT^40A;tub-GAL4*) were crossed to *w;HNF4^Δ33/ CyO twi>GFP* males for MARCM analysis. Flies were reared on the 3% sugar diet and heat treated at 37°C for one hour as pharate adults, and again as newly-eclosed adults. Adults were maintained on the 15% sugar diet for 8–10 days prior to analysis.

## Tetramethylrhodamine ethyl ester (TMRE) staining

Adult flies were dissected in room temperature Schneider's medium (ThermoFisher 21720024) with microdissection scissors to separate the abdomen from the thorax. The intestine was removed and the abdominal cavity was cut open to allow maximal contact of fat body with the TMRE solution (ThermoFisher T669). Dissected abdomens were placed in freshly-prepared solution of 20 nM TMRE in Schneider's medium and incubated in the dark for 5–7 min. Samples were then washed briefly twice in Schneider's medium, mounted on glass slides in fresh Schneider's medium, and immediately imaged on an Olympus FV1000 confocal microscope with a 60x water objective.

## Dihydroethidium (DHE) staining

DHE staining was performed as previously described (*Owusu-Ansah and Banerjee, 2009*). Briefly, adult abdomens were dissected as described for TMRE staining, but instead placed in freshly prepared 30 µM DHE (ThermoFisher D11347) in Schneider's medium and incubated in the dark for 5–7 min. Samples were washed rapidly three times with Schneider's medium, followed by fixation in 4% formaldehyde in 1xPBS for seven minutes. Samples were then washed twice in 1xPBS, mounted in SlowFade Gold (ThermoFisher S36936), and imaged on an Olympus FV1000 confocal microscope with a 20X objective.

## Statistics

GraphPad PRISM 6 software was used to plot data and perform statistical analysis. Pairwise comparison *P* values were calculated using a two-tailed Student's *t*-test, multiple comparison *P* values were calculated using one-way ANOVA with Šídák multiple test correction (except *Figure 4B,C* which included Dunnett's correction). Error bars are ± 1xSEM unless otherwise noted. Box plots display the full range of data (error bars), the 25–75th quartiles (box), and the median (midline).

## Acknowledgements

We thank J Tennessen for his contributions to the early stages of this project and for valuable discussions, P Leopold for DILP2 antibodies, S Park and S Kim for DILP2 ELISA stocks and reagents, C Doane for his contributions to the glucose tolerance test, J Evans for technical assistance, D Winge for the anti-SDHB antibodies, the University of Utah Metabolomics, Cell Imaging, High-Throughput Sequencing and Bioinformatics cores, the Bloomington Stock Center, VDRC and TRiP for stocks and reagents, and FlyBase for informatics support. This research was supported by the NIH (DK075607) and a NIH Developmental Biology Training Grant to WB (5T32 HD07491).

## Additional information

### Funding

| Funder | Grant reference number | Author |
| --- | --- | --- |
| National Institutes of Health | DK075607 | Carl S Thummel |
| National Institutes of Health | 5T32 HD07491 | William E Barry |

The funders had no role in study design, data collection and interpretation, or the decision to submit the work for publication.

### Author contributions

WEB, Conceived and designed the study, Acquired the data, Analyzed and interpreted the data, Drafted and revised the paper; CST, Conceived and designed the study, Assisted with data analysis and interpretation, Revised the paper

### Author ORCIDs

Carl S Thummel, http://orcid.org/0000-0001-8112-4643

## Additional files

### Supplementary files

• Supplementary file 1. List of genes identified by RNA-seq that display differential abundance between *dHNF4* mutant adult males and matched controls, meeting an FDR cutoff of 20 (1%) and Log2 ratio of ± 0.6 (>1.5 fold change). Transcripts that show reduced abundance in *dHNF4* mutants are marked in beige, while those with increased abundance are colored blue.

• Supplementary file 2. List of the gene ontology categories (determined using GOstat) represented in the top 500 down-regulated and 500 up-regulated genes in *dHNF4* mutant adults. The top 10-16 GO categories for each gene set are listed in order of significance along with the number of genes affected in that category, the total number of genes in that category (in parentheses), and the statistical significance of the match.

• Supplementary file 3. List of genes with transcription start sites (TSS) within 10 kb of dHNF4 enrichment peaks determined by ChIP-seq analysis, meeting an FDR 20 cutoff (1%). The chromosomal region and coordinates are indicated for each enrichment peak with neighboring genes listed below each region. Genes are listed from proximal to distal, where the distance reported represents the number of base pairs from the TSS to the middle of the enrichment peak. Gene symbols and corresponding FlyBase gene ID numbers are reported, along with chromosomal location, gene start and stop sites, strand, and TSS position. Neighboring genes also identified by RNA-seq as being either up- or down-regulated in *dHNF4* mutants are highlighted in blue and beige, respectively.

### Major datasets

The following datasets were generated:

| Author(s) | Year | Dataset title | Dataset URL | Database, license, and accessibility information |
|---|---|---|---|---|
| Barry W, Thummel CS | 2015 | RNA-seq analysis of dHNF4 mutants | http://www.ncbi.nlm.nih.gov/geo/query/acc.cgi?acc=GSE73523 | Publicly available at NCBI Gene Expression Omnibus (accession no: GSE73523) |
| Barry W, Thummel CS | 2015 | ChIP-seq analysis of dHNF4 | http://www.ncbi.nlm.nih.gov/geo/query/acc.cgi?acc=GSE73675 | Publicly available at NCBI Gene Expression Omnibus (accession no: GSE73675) |
| Thummel CS, Barry WE | 2015 | Data from: Drosophila HNF4 promotes glucose-stimulated insulin secretion and increased mitochondrial function in adults | http://dx.doi.org/10.5061/dryad.8h8q5 | Available at Dryad Digital Repository under a CC0 Public Domain Dedication |

**Reporting standards:** Standard used to collect data: Data was uploaded to the NCBI Gene Expression Omnibus website according to their specifications and guidelines.

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
