## [Decision Letter]

Thank you for submitting your work entitled "*Drosophila* HNF4 promotes glucose-stimulated insulin secretion and increased mitochondrial function in adults" for consideration by *eLife*. Your article has been favorably evaluated by K VijayRaghavan (Senior editor) and three reviewers, one of whom is a member of our Board of Reviewing Editors.

The reviewers have discussed the reviews with one another and the Reviewing editor has drafted this decision to help you prepare a revised submission.

Summary

In this manuscript the authors study the role of HNF in *Drosophila* glucose homeostasis. In humans, mutations in HNF4A cause an early onset form of diabetes. The authors make the insightful observation that mutations in HNF4 result in early adult death just after eclosion. This phenotype is rescued if the adults are fed on 3% sugar diet instead of the regular diet, which contains about 15% sugars in it. Thus, reduced carbohydrate intake increases viability of *HNF4* mutants suggesting that it might function in the regulation of sugar homeostasis. Further analysis suggests indeed that *HNF4* mutants have increased levels of free glucose and trehalose in adults and have impaired glucose tolerance and reduced levels of ATP (60-70% of the normal). The authors then use MS to determine the levels of different intermediates in glycolysis and mitochondrial metabolism and find that many of the intermediates of glycolysis and the associated pathways that function in the biosynthesis of other metabolites show a significant increase when *HNF4* mutants are fed on 15% sugar diet but not when they are reared on 3% sugar diet. Consistent with these observations the authors show that loss of HNF4 causes mitochondrial fragmentation. Based on these observations that authors conclude that HNF4 functions in the metabolic reprogramming from the larval to the adult mode during development consistent with the lethality of *HNF4* mutants.

Interestingly, mutant flies survive longer on a low sugar diet. Using global analyses such as metabolomics, RNA-seq and ChIP-seq the authors find that glycolysis and mitochondrial function are altered in mutant flies. Surprisingly, the RNA-seq and ChIP-seq data point to the possibility that HNF acts in the mitochondria to increase mtDNA transcription. The authors then use tissue specific knock-downs to show that glucose homeostasis is most perturbed when HNF is knocked down in the IPCs and fat body. Indeed, knock-down of HNF in the IPCs reduces DILP2 in circulation. The authors then explore the possibility that HNF triggers a developmentally regulated metabolic switch towards GSIS and OXPHOS at the onset of adulthood.

The authors use a range of technologies to establish links between development, metabolic regulation and glucose homeostasis. A number of novel findings are reported. The most notable (but also potentially confusing, see below) is the finding that HNF might regulate mtDNA transcription. If so, this would alter our understanding of how mitochondrial transcription is regulated, and offer insights into how mtDNA transcription might be altered in diseases, such as diabetes. Unfortunately, the experiments that this novel claim is based on are not conclusive.

The following is a list of major concerns that the reviewers have expressed and would like addressed.

Essential revisions:

1) A major, most exciting, finding of the paper is that dHNF4 might directly regulate mtDNA transcription. If so, this will change much about how we think mtDNA transcription is regulated. While the authors provide some evidence for this (for example ChIP-seq data), further experiments should be conducted to solidify this finding. In particular, it would be nice if the authors provide further evidence that dHNF4 is indeed targeted to mitochondria and present within the mitochondrial matrix where mtDNA resides. This could be achieved by subcellular fractionation of tissues or cells to show that dHNF4 first associates with the mitochondrial fraction and then further sub-fractionation to determine where within mitochondria it is. Along the same lines, are there any sequences in dHNF4 specifically required to target it to mitochondria? Does it possess a canonical N-terminal mitochondrial targeting sequence? If so, what are the phenotypic consequences of preventing mitochondrial targeting?

The key evidence that dHNF binds to mtDNA is the ChIP-seq data. It is unclear how to interpret this mtDNA Chiq-seq data for a couple of reasons:

A) The presumably high copy number of mtDNA in the sample (perhaps 1000X in excess of nuclear).

B) The low signal to noise ratio in the nuclear binding sites. Presumably, most ChIP-seq experiments will detect mtDNA as a background signal. This will usually be discarded, because of the excess signal at specific nuclear DNA binding sites. In this study, in which the signal to noise ratio in nuclear genes is quite low, any background signal caused by the mtDNA might seem more important. A nice control would have been to perform the ChIP-seq in the mutant flies. The mtDNA signal should be much reduced in the mutants (assuming the mtDNA copy number is the same in mutant flies).

2) The second piece of evidence supportive of HNF4 being present in the mitochondria is an IF experiment, which uses the same antibody as the ChIP-seq study. Given the novelty of the finding, it would have been useful to see independent validation of mitochondrial localization using an alternative antibody or a genetic construct. In summary, whilst it is plausible that HNF is directly binding to mtDNA and regulating mtDNA transcription, further evidence, using a method that doesn't require the antibody used in this study, would make the claim more compelling. Given the interest in such a finding, further independent evidence of direct mtDNA binding would be necessary.

3) As a related issue, the argument that loss of dHNF4 specifically impairs mitochondrial metabolism needs further clarification. Loss of dHNF4 could also or instead, cause a reduction in mitochondrial number and mass. To what degree can the consequences of dHNF4 depletion be attributed to a reduction of mitochondrial number or mass as opposed to specific TCA cycle and OXPHOS impairment? Does loss of dHNF4 reduce mitochondrial number, mass or mtDNA copy number?

Similarly, the mitochondrial organization is significantly impaired upon loss of HNF4 functions. Do the authors also see an increase in the expression of enzymes such as Drp1 in the RNA-seq? And does the knockdown of DRP1 function result in the rescue of the phenotype? Do ROS levels increase in mutant *HNF4* tissue and does scavenging of ROS suppress the phenotype?

Related to this issue, the authors should determine if the integrity of the cells that normally express it is compromised in any way such as an incorrect specification/physiology in the absence of HNF4, and please provide a more complete justification for why they think that the cells show a "mitochondrial fragmentation" phenotype.

For the RNA-seq data, a systematic enrichment analysis (GO term/KEGG pathway) needs to be carried out on all differentially expressed genes in order to support claims regarding specificity of pathways regulated by dHNF4. In the current version of the manuscript, the authors seem to have cherry-picked genes (the expression of which is often only mildly affected, e.g. Figure 3) to claim that dHNF4 regulates genes involved in processes such as OXPHOS.

Many of the above questions are arising, and are important to address, because the central issue has not been convincingly addressed: How does HNF4 regulate mitochondrial gene expression?

4) One potential caveat with the RNA-seq and ChIP-seq experiments is that they have been carried out in whole flies. dHNF4 is expressed in multiple tissues, and both its functions and targets may be tissue-specific. This is something that will not be captured by whole-fly approaches, and may in fact prevent identification of some targets. For example, the contribution of the IPCs to total fly chromatin or RNA is likely to be negligible.

While it is understandable that whole adult flies are used for technical reasons, presumably, each of the tissues in question will respond differently to the loss of HNF function, with different metabolic consequences, so it is unclear how to relate these global analyses of a heterogeneous tissue extract to pathology in a particular tissue (e.g. the IPCs). For example, the metabolomics data shows that HNF deletion increases glucose-derived metabolites, but the RNA-seq data suggests that glycolytic genes are down-regulated. These data are not incorrect, since the experiments and analysis were done with care and competence, but such analyses are easier to interpret if using a homogeneous extract. The ambiguities introduced by a mixture of tissues with mixed metabolic profiles make it difficult to appreciate the importance of the genomic analyses.

The Reviewing Editor does not feel that this issue can be fully solved in a reasonable period of time. So perhaps some initial experiments that address aspects of this issue, and attending to this issue in the Discussion, as a caveat will suffice.

5) All the mutants characterized in this study involve constitutive dHNF4 loss or downregulation throughout development – either from all tissues or a subset thereof. Consequently, the adult phenotypes described in this manuscript could be adaptive and result from the metabolic alterations caused by dHNF4 loss during larval life (Palanker et al. 2009). The abundance of genes involved in stress and immunity in the RNA-seq dataset is consistent with this idea. The authors should knockdown dHNF4 specifically in the adult (e.g. in tubGal4, tubGal80ts>dHNF4-RNAi flies) and test whether some of the key phenotypes (glucose intolerance, hyperglycemia and GSIS) are still apparent. Related to this, the specificity of the Ilp2-Gal4 used in this study (Kim et al. 2014) has not been investigated/described in larvae. Why did the authors not use the Ilp2-3-Gal4 generated in the Partridge lab, which is confined to late larva/adult IPCS? This would avoid targeting non-IPC dHNF4 expression throughout development.

Also, a related important but minor issue is that often a single RNAi construct is used, the authors are well aware of the possible pitfalls, and they need to be avoided. Additionally, all UAS and Gal4 lines need to be fully defined.

6) The authors argue that dHNF4 is required in the fat body and insulin-producing cells of adults to maintain glucose homeostasis by supporting a developmental switch toward OXPHOS and GSIS at the transition to adulthood. Since all experiments in this paper were performed by removing dHNF4 early in development far before the onset of adulthood, it remains a possibility that the phenotypes observed are a consequence of an earlier requirement for dHNF4 during development. To show that dHNF4 is really required in adulthood we would like the authors to remove dHNF4 at the onset of adulthood in the whole organism, fat body and IPCs to assess whether the flies become diabetic and whether survival is affected. As the muscle is also a major contributor to glucose clearance the requirement of dHNF4 in muscles should be assessed as well.

7) Based on RNAi to knock-down of specific OXPHOS complexes in the IPC and fat body the authors conclude that there are differential requirements for different ETC complexes in glucose homeostasis. This requires functional validation e.g. on the rate of oxygen consumption. The authors demonstrate that *Cox5a* knock-down does not increase glucose, but CIA30 knock-down does. It is more likely that each complex has differential control of flux in specific tissues or flux through the ETC has differential sensitivity to knock-down of specific complexes, rather than suggesting that there is a greater requirement for CI or CV than CIV per se.

8) The authors' previous study was consistent with dHNF4 as a nutrient sensor, whereas this manuscript suggests it functions as a developmental switch that converts a cell to a fate that supports responsiveness to glucose. Are dHNF4 expression, activity and/or subcellular localization nutritionally regulated in adult flies? The very low number of ChIP-seq could be explained low dHNF4 activity if a 3% sugar diet was used. RNA-seq experiments used 15% sugar. The diets used for every experiment also need to be clearly stated, and the data analyzed accordingly.

---

## [Author Response]

Essential revisions:

1) A major, most exciting, finding of the paper is that dHNF4 might directly regulate mtDNA transcription. If so, this will change much about how we think mtDNA transcription is regulated. While the authors provide some evidence for this (for example ChIP-seq data), further experiments should be conducted to solidify this finding. In particular, it would be nice if the authors provide further evidence that dHNF4 is indeed targeted to mitochondria and present within the mitochondrial matrix where mtDNA resides. This could be achieved by subcellular fractionation of tissues or cells to show that dHNF4 first associates with the mitochondrial fraction and then further sub-fractionation to determine where within mitochondria it is. Along the same lines, are there any sequences in dHNF4 specifically required to target it to mitochondria? Does it possess a canonical N-terminal mitochondrial targeting sequence? If so, what are the phenotypic consequences of preventing mitochondrial targeting?

We agree with the reviewers that the discovery that dHNF4 may directly regulate mitochondrial gene expression and function is an exciting part of our paper. The evidence in support of this model includes (1) reduced expression of most of the 13 protein-coding mitochondrial genes in *dHNF4* mutants. (2) Localization of dHNF4 protein to the predicted mtDNA promoter region by ChIP-seq, (3) no apparent change in mitochondrial DNA number (which we now show in Figure 3) and no effect on mitochondrial *mt:Cyt-b* transcription, which is predicted to be expressed from an independent mtDNA promoter, (4) localization of dHNF4 protein to both nuclei and mitochondria in antibody stains, and (5) indicators of mitochondrial dysfunction, including metabolomic data, reduced ATP levels, and mitochondrial fragmentation. We also now include new data showing that our antibody localization of dHNF4 to mitochondria is specific by showing that this staining is absent in *dHNF4* mutant tissue (Figure 3—figure supplement 2). In addition, we show that the staining for mitochondrial dHNF4 is punctate, resembling that reported for mtDNA stains within mitochondria (Figure 3). We also show that mitochondrial membrane potential is significantly reduced in *dHNF4* mutants (Figure 3—figure supplement 2), consistent with the reduced ATP levels and fragmentation phenotype (Figure 1, and Figure 3—figure supplement 2).

The reviewers ask if dHNF4 has a canonical N-terminal mitochondrial targeting sequence. This is an important question and, indeed, *dHNF4* does encode such a protein. We have surveyed the protein isoforms encoded by all 18 canonical nuclear receptors in *Drosophila* and have found only two that carry a N-terminal mitochondrial targeting signal (MTS) predicted by both MITOPROT and TargetP protein localization programs. Importantly, one of these is dHNF4-B, one of the five protein isoforms encoded by the *dHNF4* locus (as we now depict in Figure 3—figure supplement 3). We are using CRISPR/Cas9 to specifically delete the 5’ exon that is unique to *dHNF4-B*, leaving the remaining four isoforms intact. We are also establishing a transgenic line that expresses a tagged dHNF4-B protein under UAS control. Both of these constructs are currently being injected into embryos. We are also engineering an N-terminal protein tag into the dHNF4-B-specific coding sequences to facilitate its detection, along with lines that will carry a protein tag at the C-terminus of the dHNF4 protein shared by all five isoforms. Our plans are to pursue a complete analysis of these new lines, along with our existing *dHNF4* mutant, to dissect the relative contributions of nuclear and mitochondrial function to *dHNF4* mutant phenotypes. In addition, we have initiated collaborative efforts with colleagues at the University of Utah to determine if the mitochondrial localization and function of HNF4 is conserved through evolution from flies to mammals. Taken together, these studies represent a major direction for our future research. An NIH R01 grant to support this project recently received a good score and is being resubmitted. More relevant to this paper, these detailed studies of potential functions for dHNF4 in mitochondria extend beyond what we hope to communicate in this paper, which represents a much broader initial phenotypic characterization of the adult functions for *dHNF4* in *Drosophila*.

We fully agree with the reviewers that mitochondrial fractionation is important to both confirm our model and determine the location of dHNF4 within mitochondria. Indeed, we were doing these experiments while the paper was out for review and have continued our efforts over the past few months. So far, we have conducted six attempts using control animals compared to *dHNF4* mutants as well as a genomic rescue transgene for dHNF4 tagged with GFP and FLAG epitope tags. These efforts, however, have been plagued by a high background of non-specific bands in mitochondrial fractions by western blot that obscure any potential signal from dHNF4. Surprisingly, these non-specific bands are seen in the highly purified mitochondrial fractions (as determined by the fractionation of specific marker proteins) but not in isolated nuclei. We have conducted these studies with a variety of different antibodies including affinity-purified guinea pig anti-dHNF4 (unpublished from our lab), affinity-purified rat anti-dHNF4 (Palankeret al., 2009), chicken anti-GFP (Abcam 13970), and mouse anti-FLAG monoclonal (Sigma F3165). We also attempted pre-absorbing the antibodies using purified mitochondria, but this was not sufficient to fully eliminate interference from non-specific bands. Our need for additional approaches to detect dHNF4 protein has been a major motivating factor for developing the new reagents described above that tag all dHNF4 protein isoforms as well as the dHNF4-B isoform. We now include in our paper a discussion of the importance of conducting mitochondrial fractionation studies in the future (subsection “dHNF4 appears to directly regulate mitochondrial gene expression and function”, third paragraph).

The focus of the reviewers on requesting additional evidence for the mitochondrial functions of dHNF4 is our fault, due to a vague discussion of this topic throughout the paper. At no point in our paper did we enumerate the evidence for and against a direct role for dHNF4 in mitochondria, as well as the key experiments that are needed to conclusively prove this model. These include a full functional analysis of the dHNF4-B isoform and its contributions to *dHNF4* mutant phenotypes. We now include this information in a new section of our Discussion that is focused on mitochondria, along with our conclusion that, although our data strongly support the model that dHNF4 directly regulates mitochondrial gene expression, they do not prove it. We now present this as a working model. We have revised all parts of the text that previously referred to “direct regulation of mitochondrial gene expression” to be clear that we cannot make this conclusion based on the evidence presented. We also now state that substantial future studies will be needed to provide conclusive evidence in support of our model. We invite the reviewers to read this modified section of the Discussion (“dHNF4 appears to directly regulate mitochondrial gene expression and function”).

Finally, we want to point out that, although the reviewers are focused on the mitochondrial functions for dHNF4, this is only one of three major findings in our paper. Importantly, this study provides the first accurate animal model for MODY1 and thus the first insights into the molecular mechanisms that underlie this disorder. In addition, we show that the induction of dHNF4 in early adults provides a regulatory mechanism to explain the classic “U-shaped curve” of mitochondrial oxygen consumption in holometabolous insects – consisting of high respiration rates in larvae, low levels during metamorphosis, and a dramatic increase at the onset of adulthood. Our data supports the model that dHNF4 contributes to establishing the metabolic state of the adult fly by supporting both GSIS and mitochondrial activity. This coupling of developmental progression with metabolic state is a critical regulatory connection that is poorly understood in all animals. In addition, our paper defines a developmental shift in systemic glucose homeostasis and provides new evidence to explain why adult IPCs secrete DILP2 in response to glucose, unlike those of larvae. Finally, it is important to note that this is the first paper to report functional studies of *dHNF4* in adults. We agree that many experiments can and should be done – particularly regarding its roles in mitochondria – and it is our intention to conduct these experiments over the coming years. We have rewritten major portions of the manuscript to be sure that these points are made clear to the reader. We have also added 7 new figures to the revised manuscript, significantly expanding the scope of our study.

The key evidence that dHNF binds to mtDNA is the ChIP-seq data. It is unclear how to interpret this mtDNA Chiq-seq data for a couple of reasons:

A) The presumably high copy number of mtDNA in the sample (perhaps 1000X in excess of nuclear).

*B) The low signal to noise ratio in the nuclear binding sites. Presumably, most ChIP-seq experiments will detect mtDNA as a background signal. This will usually be discarded, because of the excess signal at specific nuclear DNA binding sites. In this study, in which the signal to noise ratio in nuclear genes is quite low, any background signal caused by the mtDNA might seem more important. A nice control would have been to perform the ChIP-seq in the mutant flies. The mtDNA signal should be much reduced in the mutants (assuming the mtDNA copy number is the same in mutant flies).* The reviewers are correct that the abundance of mtDNA can complicate ChIP-seq peak calling for the mitochondrial genome. Indeed, we were aware of this and thus our analysis of the ChIP-seq data was performed using the MACS2 platform, which addresses this potential artifact by accounting for local differences in read depth across the genome (including the increased abundance of mtDNA relative to nuclear DNA). This program performs peak calling by calculating dynamic Poisson distribution parameters (local lambda values) and thus adjusts for biases due to local differences in abundance compared to the rest of the genome. In effect, this results in peak calling relative to protein binding levels within adjacent flanking sequences of DNA. As a result, we detect specific dHNF4 enrichment that is restricted to a portion of the mtDNA control region and, importantly, this enrichment is not observed in the input control. Although the triple-stranded structure of the D-loop in the control region of mammalian mtDNA has been proposed to contribute to possible false-positive ChIP-seq peaks in mammalian datasets (Marinov et al., 2014), the D-loop structure is not present in *Drosophila* mtDNA (Rubensteinet al., 1977). Furthermore, the peak for dHNF4 on mtDNA represents the 5^th^ strongest enrichment peak in our ChIP-seq dataset out of 68 total. This suggests that its significance is not overemphasized due to low signal, as the high-confidence direct targets for dHNF4 identified from this analysis show robust enrichment for proximal dHNF4 binding. In addition, only select mtDNA genes are underexpressed in *dHNF4* mutants while mtDNA copy number is unaffected, suggesting a specific regulatory defect in mitochondrial gene expression. Finally, in two other ChIP-seq studies performed in our lab for other nuclear receptors in adult flies using the same protocol, we have not observed such localized enrichment to this predicted mtDNA promoter region. As a result, we feel that specific binding of dHNF4 to mtDNA is a reasonable conclusion from our studies. We have added the information cited above to our Discussion in an effort to address this important concern for readers (subsection “dHNF4 appears to directly regulate mitochondrial gene expression and function”).

Although we agree that it would be ideal to perform a ChIP-seq study in *dHNF4* mutant flies as the reviewers request, this is an expensive experiment to conduct as a control for the specificity of dHNF4 binding to mtDNA. Indeed, we know of no comparable control studies in the literature. In addition, the low viability of *dHNF4* mutants makes it difficult to get the large quantities of fresh material we need for this experiment (gram quantities of flies). We have attempted to design PCR primers to perform focused ChIP on mtDNA, followed by qPCR, but the sequences within and proximal to the region of dHNF4 binding lack suitable annealing sites due to AT-rich content.

*2) The second piece of evidence supportive of HNF4 being present in the mitochondria is an IF experiment, which uses the same antibody as the ChIP-seq study. Given the novelty of the finding, it would have been useful to see independent validation of mitochondrial localization using an alternative antibody or a genetic construct. In summary, whilst it is plausible that HNF is directly binding to mtDNA and regulating mtDNA transcription, further evidence, using a method that doesn't require the antibody used in this study, would make the claim more compelling. Given the interest in such a finding, further independent evidence of direct mtDNA binding would be necessary.* We fully agree with the reviewers that additional evidence is needed to support our detection of dHNF4 in mitochondria by IF. Accordingly, we now include antibody stains of adult fat body from both controls and *dHNF4* mutants, conducted in parallel and imaged under identical conditions. This data, included as Figure 3—figure supplement 2, shows that the mitochondrial signal for dHNF4 is absent in mutants. We hope that this new data convincingly demonstrates the specificity of our antibody for immunofluorescent studies.

In addition, we agree that our IF studies would benefit from using a second independent antibody to detect dHNF4. Unfortunately, however, no such reagent exists. In our original work, we raised antibodies against a 317 amino acid region shared by all dHNF4 protein isoforms, using both rats and guinea pigs. Both rodent antibodies were affinity-purified, and both detect similar patterns of cytoplasmic dHNF4 protein. The rat antibody, however, has a significantly higher signal to noise ratio as determined by stains of mutant tissue. We use this antibody for all our published work. In an effort to address the reviewers concerns, we obtained the GFP+FLAG-tagged dHNF4 BAC transgenic line from Bloomington (stock 38649). However, efforts to detect either GFP or FLAG tags using several antibodies (chicken anti-GFP Abcam 13970, rabbit anti-GFP MBL 598, rabbit anti-FLAG Σ F7425) resulted in significant background signal in the cytoplasm from cells lacking this transgene. As stated above, we are currently developing multiple tagged lines with the express purpose of addressing this important issue raised by the reviewers as part of our future studies.

*3) As a related issue, the argument that loss of dHNF4 specifically impairs mitochondrial metabolism needs further clarification. Loss of dHNF4 could also or instead, cause a reduction in mitochondrial number and mass. To what degree can the consequences of dHNF4 depletion be attributed to a reduction of mitochondrial number or mass as opposed to specific TCA cycle and OXPHOS impairment? Does loss of dHNF4 reduce mitochondrial number, mass or mtDNA copy number?* This is an important question. We agree that this is a critical experiment, which is now included as Figure 3. This data shows that there is no significant effect on mitochondrial number as reflected by the ratio of mtDNA:nuclear DNA, the conventional measurement for mitochondrial number. Furthermore, we performed additional metabolomics analysis by GC/MS under conditions where amino acids are restricted from the diet (sucrose fed for 3 days) to limit their ability to replenish TCA cycle intermediates through anapleurotic reactions, providing a more robust approach for identifying changes in TCA cycle intermediates. These studies demonstrate specific defects in the TCA cycle that correlate with the changes in TCA cycle gene expression observed in *dHNF4* mutants. This new data is presented as Figure 3—figure supplement 1 and discussed in the second paragraph of the subsection “*dHNF4* mutants display defects in glycolysis and mitochondrial metabolism”. We also show reduced levels of Complex II by western blot, consistent with the reduced expression of the critical Complex II assembly factor gene *dSdhaf4* in *dHNF4* mutants (Figure 3—figure supplement 2). Together, these data indicate that *dHNF4* mutants exhibit specific defects in mitochondrial metabolism and gene expression that cannot be explained by an overall reduction in mitochondrial number.

*Similarly, the mitochondrial organization is significantly impaired upon loss of HNF4 functions. Do the authors also see an increase in the expression of enzymes such as Drp1 in the RNA-seq? And does the knockdown of DRP1 function result in the rescue of the phenotype? Do ROS levels increase in mutant HNF4 tissue and does scavenging of ROS suppress the phenotype?* These are excellent questions that begin to address the causes of mitochondrial fragmentation that we observe in *dHNF4* mutants. We have scanned our list of dHNF4-regulated genes ([Supplementary-material SD1-data]) for Drp1 (CG3210), as requested, as well as Opa1 (CG8479), Mfn1/2 (CG3869), Fis1 (CG17510), and Fzo (CG4568). None of these regulators of mitochondrial dynamics, however, are affected by the *dHNF4* mutation. Rather than being a direct response to the misregulation of a dHNF4 target gene, however, the mitochondrial fragmentation that we observe in *dHNF4* mutants is more likely due to their overall defects in mitochondrial activity. This association is well established in the literature, as we now discuss in the subsections “dHNF4 acts through multiple tissues and pathways to control glucose homeostasis “and “dHNF4 appears to directly regulate mitochondrial gene expression and function”. Consistent with this, disruption of the electron transport chain via RNAi knockdown of the Complex I assembly factor, *CIA30*, or knockdown of *CG5599* (a direct target of dHNF4 that encodes a protein with homology to the E2 subunit of α-ketoglutarate dehydrogenase), in the fat body results in mitochondrial fragmentation and hyperglycemia, similar to loss of dHNF4. This data is now included in Figure 4. In addition, we now include a clonal analysis experiment, depicted in Figure 3—figure supplement 2, which shows reduced TMRE staining in clonal patches of *dHNF4* mutant cells adjacent to wild-type cells – indicative of reduced mitochondrial membrane potential. This result is consistent with the reduced ATP levels that we detect in *dHNF4* mutants and provides a mechanism that can explain the mitochondrial fragmentation, as we now include in the Discussion (in the aforementioned subsection). We have also assayed for changes in ROS as requested by the reviewers. This is shown in Figure 3—figure supplement 2. We do not, however, see a significant change in ROS by this assay, in either the intestine or fat body. This is consistent with our inability to rescue *dHNF4* mutant phenotypes by feeding the animals N-acetyl-cysteine, which provides effective antioxidant activity in *Drosophila*. One possible explanation for this is that the ETC complexes that produce ROS are expressed at reduced levels in *dHNF4* mutants, while the ROS-scavenging enzymes, such as catalase and SOD, remain unaffected in these animals. We now comment on this in the paper (subsection “dHNF4 regulates nuclear and mitochondrial gene expression”, last paragraph).

*Related to this issue, the authors should determine if the integrity of the cells that normally express it is compromised in any way such as an incorrect specification/physiology in the absence of HNF4, and please provide a more complete justification for why they think that the cells show a "mitochondrial fragmentation" phenotype.* We see no changes in the size or morphology of any cells or tissues we have examined in *dHNF4* mutants, with a particular focus on the IPCs (which are central to MODY in humans). The mitochondrial fragmentation that we see in *dHNF4* mutants or upon *dHNF4* RNAi, depicted in Figure 4 and Figure 3—figure supplement 2, is one of our most penetrant phenotypes, and this is a common response to mitochondrial dysfunction that has been widely observed and studied. Consistent with this, we now show that fat body-specific RNAi for the dHNF4 target gene, *CG5599*, which encodes a protein with homology to α-ketoglutarate dehydrogenase, or disruption of the ETC by RNAi against *CIA30*, results in mitochondrial fragmentation that resembles that seen in *dHNF4* mutant tissue (Figure 4). In addition, we show that mitochondrial morphology is normal upon *Hex-C* RNAi indicating that fragmentation is not a nonspecific effect or a secondary response to hyperglycemia. We comment on this fragmentation in the context of the reduced mitochondrial membrane potential and ATP levels in the mutants, and cite appropriate references in the Discussion (subsections “dHNF4 acts through multiple tissues and pathways to control glucose homeostasis “and “dHNF4 appears to directly regulate mitochondrial gene expression and function”).

*For the RNA-seq data, a systematic enrichment analysis (GO term/KEGG pathway) needs to be carried out on all differentially expressed genes in order to support claims regarding specificity of pathways regulated by dHNF4. In the current version of the manuscript, the authors seem to have cherry-picked genes (the expression of which is often only mildly affected, e.g. Figure 3) to claim that dHNF4 regulates genes involved in processes such as OXPHOS.* We thank the reviewers for this suggestion. We now include a GO analysis in [Supplementary-material SD2-data]. This lists the top 500 up- and down-regulated genes in our RNA-seq analysis of *dHNF4* mutants. For the down-regulated genes, the top category is oxidoreductase, which includes many mitochondrial proteins that are involved in redox reactions (p=3.1e-41). The next most significant categories primarily represent genes involved in metabolism, with “electron carrier activity” as number 4. For the up-regulated genes, all of the top categories are involved in antibacterial and innate immune responses. We now comment on this analysis in the subsections “dHNF4 regulates nuclear and mitochondrial gene expression“, first paragraph, and “dHNF4 acts through multiple pathways to regulate glucose homeostasis”, fourth paragraph.

*Many of the above questions are arising, and are important to address, because the central issue has not been convincingly addressed: How does HNF4 regulate mitochondrial gene expression?* As outlined in our response to comment #1, we fully agree that significant further studies are needed to definitively demonstrate that dHNF4 directly regulates mitochondrial gene expression and function. We wish to emphasize, however, that our current paper is not intended to focus on this topic. Rather, it represents the first phenotypic study of *dHNF4* mutant adults and a functional analysis of the roles for dHNF4 in maintaining carbohydrate homeostasis. In addition to providing compelling evidence for a direct mitochondrial function for dHNF4, our paper provides the first animal model for MODY1, 20 years after the discovery of this genetic association in humans. Our paper also establishes an important role for *dHNF4* in contributing to a developmental switch in metabolic state and provides a molecular basis to understand the U-shaped curve of mitochondrial activity. We hope that the reviewers will agree that this work, taken together, is worthy of publication in *eLife*.

4) One potential caveat with the RNA-seq and ChIP-seq experiments is that they have been carried out in whole flies. dHNF4 is expressed in multiple tissues, and both its functions and targets may be tissue-specific. This is something that will not be captured by whole-fly approaches, and may in fact prevent identification of some targets. For example, the contribution of the IPCs to total fly chromatin or RNA is likely to be negligible.

*While it is understandable that whole adult flies are used for technical reasons, presumably, each of the tissues in question will respond differently to the loss of HNF function, with different metabolic consequences, so it is unclear how to relate these global analyses of a heterogeneous tissue extract to pathology in a particular tissue (e.g. the IPCs). For example, the metabolomics data shows that HNF deletion increases glucose-derived metabolites, but the RNA-seq data suggests that glycolytic genes are down-regulated. These data are not incorrect, since the experiments and analysis were done with care and competence, but such analyses are easier to interpret if using a homogeneous extract. The ambiguities introduced by a mixture of tissues with mixed metabolic profiles make it difficult to appreciate the importance of the genomic analyses. The Reviewing Editor does not feel that this issue can be fully solved in a reasonable period of time. So perhaps some initial experiments that address aspects of this issue, and attending to this issue in the Discussion, as a caveat will suffice.* The reviewers make the important point that different tissues can respond in different ways to a loss-of-function mutation. Tissue-specific RNA-seq, ChIP-seq, and GC/MS metabolomic analysis would address this issue, but they require significant technical effort and financial resources. Rather, we have chosen to address this topic through tissue-specific functional studies of select target genes for dHNF4, as depicted in Figure 4 and 5. To address this important topic, we now discuss the caveats involved in interpreting whole-animal genomewide and metabolomic studies in the Discussion, subsection “dHNF4 acts through multiple pathways to regulate glucose homeostasis”, second paragraph.

5) All the mutants characterized in this study involve constitutive dHNF4 loss or downregulation throughout development – either from all tissues or a subset thereof. Consequently, the adult phenotypes described in this manuscript could be adaptive and result from the metabolic alterations caused by dHNF4 loss during larval life (Palanker et al. 2009). The abundance of genes involved in stress and immunity in the RNA-seq dataset is consistent with this idea. The authors should knockdown dHNF4 specifically in the adult (e.g. in tubGal4, tubGal80ts>dHNF4-RNAi flies) and test whether some of the key phenotypes (glucose intolerance, hyperglycemia and GSIS) are still apparent.

We agree that it is important to selectively remove dHNF4 function from adults and then conduct a complete phenotypic analysis to determine how the resulting phenotypes compare to those of the germline mutant used in our paper. Accordingly, we established the recommended *tubGal80ts>dHNF4-RNAi* stock and tested this extensively in adults. Unfortunately, however, we could not obtain a sufficient reduction in *dHNF4* expression to result in reproducible phenotypes. As an alternative approach, we have used CRISPR/Cas9 to engineer two FRT sites into introns in the endogenous *dHNF4* locus in an effort to generate a conditional allele that lacks all *dHNF4* functions. We believe that this is a better way to conduct this experiment, allowing us to selectively eliminate dHNF4 function in a temporally and spatially regulated manner. We have successfully used this approach to conduct adult-specific functional studies of another nuclear receptor, dERR, in our lab. We thus feel confident that this approach will allow us to perform a comprehensive adult-specific functional analysis in the future. To address this concern of the reviewers, we now discuss this topic in the Discussion (fifth paragraph of the subsection “dHNF4 acts through multiple pathways to regulate glucose homeostasis”), pointing out that developmental functions for dHNF4 could contribute to the phenotypes we observe.

*Related to this, the specificity of the Ilp2-Gal4 used in this study (Kim et al. 2014) has not been investigated/described in larvae. Why did the authors not use the Ilp2-3-Gal4 generated in the Partridge lab, which is confined to late larva/adult IPCS? This would avoid targeting non-IPC dHNF4 expression throughout development.* We agree with the reviewers that restricting RNAi against *dHNF4* to the adult IPCs would be ideal to avoid non-specific effects from such constructs being expressed during larval development. The *dilp2-GAL4* line we use is recombined into a complex genetic background developed by Seung Kim’s lab to allow for picomolar detection of circulating DILP2 peptide (DILP2HF) upon targeted RNAi in these cells. For details, see our Materials and methods and the paper from the Kim lab (Park et al., 2014). We wanted to ensure that all of our functional studies could be related to the critical phenotype of glucose-stimulated DILP secretion, given that this is a hallmark for MODY1. We thus used this same *dilp2-GAL4* driver for all functional studies in our paper. Importantly, this line has been validated to drive expression specifically in adult IPCs (Park et al., 2014). In addition, this line includes a *UAS-Dcr* transgene in the background, which we have found is critical for efficient knockdown in IPCs. Finally, it is important to note that dHNF4 is not detectable in the IPCs during larval stages (Palanker et al., 2009) (Figure 6), and we express mCherry RNAi for all our studies to control for potential non-specific effects of RNAi in cells lacking an endogenous (or expressed) target mRNA during development.

The reviewers ask about a Ilp2-3-GAL4 line generated by the Partridge lab. To our knowledge, however, this line does not exist. Perhaps the reviewers are referring to the Ilp2-GAL4 line generated by the Hafen lab (Ilp215-3-GAL4), which has been misattributed to the Partridge lab in at least one previous publication. The Ilp215-3-GAL4 driver, however, has been reported to initiate expression during the late L3 and thus would not provide the desired adult-specific expression (Ikeya et al., 2002). In addition, there is no data to support this temporal induction since, to our knowledge, a systematic time course study of Ilp215-3-GAL4 expression has not been published.

*Also, a related important but minor issue is that often a single RNAi construct is used, the authors are well aware of the possible pitfalls, and they need to be avoided. Additionally, all UAS and Gal4 lines need to be fully defined.* We thank the reviewers for this suggestion and fully acknowledge the limitations of RNAi, especially when only a single RNAi line is available. We have used multiple RNAi lines whenever possible, in particular for our most critical results, focusing on hyperglycemia and the defects in GSIS. We use two separate lines for *dHNF4* RNAi and examine the effects on glucose levels and DILP2 secretion (Figure 4, Figure 4—figure supplement 1, Figure 5, and Figure 5—figure supplement 1). We also now include data showing our use of three separate lines for *Hex-C* RNAi (Figure 4, Figure 4—figure supplement 3). These data all show similar increases in glucose levels in the *Hex-C* RNAi animals. Additionally, strain-specific identification information for all UAS and GAL4 lines is included in both the Materials and methods section (subsection “GAL4/UAS lines for tissue-specific RNAi studies”) and within the relevant figure legends.

*6) The authors argue that dHNF4 is required in the fat body and insulin-producing cells of adults to maintain glucose homeostasis by supporting a developmental switch toward OXPHOS and GSIS at the transition to adulthood. Since all experiments in this paper were performed by removing dHNF4 early in development far before the onset of adulthood, it remains a possibility that the phenotypes observed are a consequence of an earlier requirement for dHNF4 during development. To show that dHNF4 is really required in adulthood we would like the authors to remove dHNF4 at the onset of adulthood in the whole organism, fat body and IPCs to assess whether the flies become diabetic and whether survival is affected. As the muscle is also a major contributor to glucose clearance the requirement of dHNF4 in muscles should be assessed as well.* Please see the first section of #5 above for our response to this request to perform adult-specific loss-of-function studies to complement our studies using germline *dHNF4* null mutants.

*7) Based on RNAi to knock-down of specific OXPHOS complexes in the IPC and fat body the authors conclude that there are differential requirements for different ETC complexes in glucose homeostasis. This requires functional validation e.g. on the rate of oxygen consumption. The authors demonstrate that Cox5a knock-down does not increase glucose, but CIA30 knock-down does. It is more likely that each complex has differential control of flux in specific tissues or flux through the ETC has differential sensitivity to knock-down of specific complexes, rather than suggesting that there is a greater requirement for CI or CV than CIV per se.* We thank the reviewers for raising this point. We have removed this speculation from the text since we cannot be sure of the effect of each RNAi knockdown on ETC activity, as the reviewers point out.

8) The authors' previous study was consistent with dHNF4 as a nutrient sensor, whereas this manuscript suggests it functions as a developmental switch that converts a cell to a fate that supports responsiveness to glucose. Are dHNF4 expression, activity and/or subcellular localization nutritionally regulated in adult flies? The very low number of ChIP-seq could be explained low dHNF4 activity if a 3% sugar diet was used. RNA-seq experiments used 15% sugar. The diets used for every experiment also need to be clearly stated, and the data analyzed accordingly.

The reviewers raise the interesting and important possibility that dHNF4 expression, activity and/or subcellular localization might be regulated by nutrition in *Drosophila*. Needless to say, we wondered the same thing and have conducted a number of nutritional studies, with a particular focus on changes in the subcellular localization of dHNF4. We could not, however, detect any reproducible differences. The levels of *dHNF4* mRNA are also unaffected by fasting and starvation, as demonstrated by both larval and adult microarray studies done in our lab, and several studies done by other labs. With regard to the diet used for the ChIP-seq study, we used a normal sugar diet since this experiment was conducted using wild-type (more accurately, w[1118]) *Drosophila* adults. We now clearly state the diets for each experiment in the legends and within the Materials and methods section.